# The sports nutrition knowledge of large language model (LLM) artificial intelligence (AI) chatbots: An assessment of accuracy, completeness, clarity, quality of evidence, and test-retest reliability

Thomas P. J. Solomon[1], Matthew J. Laye[2]*

**1** Blazon Scientific, London, United Kingdom, **2** Idaho College of Osteopathic Medicine, Meridian, Idaho, United States of America

* mlaye@icom.edu

## Abstract

### Background

Generative artificial intelligence (AI) chatbots are increasingly utilised in various domains, including sports nutrition. Despite their growing popularity, there is limited evidence on the accuracy, completeness, clarity, evidence quality, and test-retest reliability of AI-generated sports nutrition advice. This study evaluates the performance of ChatGPT, Gemini, and Claude's basic and advanced models across these metrics to determine their utility in providing sports nutrition information.

### Materials and methods

Two experiments were conducted. In Experiment 1, chatbots were tested with simple and detailed prompts in two domains: *Sports nutrition for training* and *Sports nutrition for racing*. Intraclass correlation coefficient (ICC) was used to assess interrater agreement and chatbot performance was assessed by measuring accuracy, completeness, clarity, evidence quality, and test-retest reliability. In Experiment 2, chatbot performance was evaluated by measuring the accuracy and test-retest reliability of chatbots' answers to multiple-choice questions based on a sports nutrition certification exam. ANOVAs and logistic mixed models were used to analyse chatbot performance.

### Results

In Experiment 1, interrater agreement was good (ICC = 0.893) and accuracy varied from 74% (Gemini1.5pro) to 31% (ClaudePro). Detailed prompts improved Claude's accuracy but had little impact on ChatGPT or Gemini. Completeness scores were

**Data availability statement:** The datasets supporting the conclusions of this article are available in the Open Science Foundation (OSF) data registry at https://osf.io/k2c6t/. The code used for statistical analyses is available on GitHub (https://github.com/tpjsolomon/AI_in_sports_nutrition) [74].

**Funding:** The work was funded by internal institutional funds from the Idaho College of Osteopathic Medicine. No external funds were received for this work. The funders had no role in study design, data collection and analysis, decision to publish, or preparation of the manuscript.

**Competing interests:** TS has given invited talks at societal conferences and university/pharmaceutical symposia for which the organisers paid for travel and accommodation; he has also received research money from publicly funded national research councils and medical charities, and private companies, including Novo Nordisk Foundation, AstraZeneca, Amylin, AP Møller Foundation, and Augustinus Foundation; and, he has consulted for Boost Treadmills, GU Energy, and Examine.com, and owns a consulting business, Blazon Scientific, and an endurance athlete education business, Veohtu. These companies have had no control over the research design, data analysis, or publication outcomes of this work. ML has given invited talks at societal conferences and university symposia and meetings for which the organisers paid for travel and accommodation; he has received research money from Augustinus Foundation, American College of Sports Medicine, and national research institutions; and, he has consulted for Zepp Health, Levels Health, GU Energy, and EAB labs, and has coached for Sharman Ultra Coaching. These companies have had no control over the research design, data analysis, or publication outcomes of this work. My Sports Dietitian provided a set of multiple-choice questions designed to resemble the Certified Specialist in Sports Dietetics (CSSD) board exam. Neither TPJS nor MJL have any financial relationships with My Sports Dietitian. This does not alter our adherence to PLOS ONE policies on sharing data and materials.

highest for ChatGPT-4o compared to other chatbots, which scored low to moderate. The quality of cited evidence was low for all chatbots when simple prompts were used but improved with detailed prompts. In Experiment 2, accuracy ranged from 89% (Claude3.5Sonnet) to 61% (ClaudePro). Test-retest reliability was acceptable across all metrics in both experiments.

## Conclusions

While generative AI chatbots demonstrate potential in providing sports nutrition guidance, their accuracy is moderate at best and inconsistent between models. Until significant advancements are made, athletes and coaches should consult registered dietitians for tailored nutrition advice.

---

## Introduction

The popularity of generative artificial intelligence (AI) chatbots has grown significantly [1,2] and several large language models (LLMs) have been developed and released as freely accessible generative AI chatbots. These include ChatGPT (OpenAI), Gemini (Google), Copilot (Microsoft), and Claude (Anthropic). Similarly, various AI-driven fitness coaching [3–9] and diet/nutrition apps [10–13] have emerged to assist athletes in creating training plans and managing their body composition.

Optimal sports nutrition practices play a crucial role in enhancing training adaptations necessary for improving endurance athletes' performance [14–16]. While the validity of generative AI chatbots in prescribing training [17–19] and diet plans [20–24] has been investigated, most studies have employed qualitative methods [18–21]. Additionally, few studies have directly compared the accuracy of different chatbots [17,18,23] and the completeness of information, clarity of outputs, and evidence quality have rarely been assessed [25–27]. For athletes to effectively use chatbots to inform their sports nutrition practices, the information provided must be accurate, complete, clear, and based on high-quality evidence. However, the current understanding of the quality of sports nutrition information generated by AI chatbots is incomplete.

Although AI chatbots can respond to simple single-sentence prompts (zero-shot prompts), the best results are typically achieved using more detailed prompts designed with prompt engineering principles, such as multi-shot or chain-of-thought prompting [28–35]. Nonetheless, chatbots currently lack built-in fact-checking capabilities and do not provide accuracy or uncertainty metrics. Furthermore, chatbot outputs are influenced by several biases, including training data bias, cultural/societal bias, confirmation bias, and recency bias [36]. Therefore, regardless of the prompt's complexity, the quality of chatbot outputs must be critically evaluated. Moreover, reliable chatbot performance requires consistent responses to identical prompts, yet AI chatbots can produce variable outputs upon repeated prompting [26,37–40]. Notably, the test-retest reliability of sports nutrition information generated by AI chatbots has not been systematically studied. While some research has found that chatbots can

outperform both the general population and ultra-endurance athletes in sports nutrition knowledge [41], comprehensive evaluations of chatbot performance across multiple metrics — accuracy, completeness, clarity, evidence quality, etc. — remain limited.

Given the uncertain accuracy, completeness, clarity, evidence quality, and test-retest reliability of AI chatbots in sports nutrition, and considering the widespread availability of multiple generative AI chatbots, it is essential to assess their performance thoroughly. Consequently, this study evaluated whether publicly available chatbots powered by different LLMs could provide high-quality sports nutrition information. Specifically, the study compared ChatGPT, Gemini, and Claude in terms of accuracy (i.e., correct answers), completeness (i.e., providing all possible correct answers), clarity (i.e., clear and fully understandable answers), evidence quality, and test-retest reliability when offering nutritional guidance that would enhance an endurance athlete's training adaptations and race day performance. Additionally, the study examined these chatbots' accuracy and reliability in answering sports nutrition certification exam questions. It was hypothesized that AI chatbots could deliver high-quality sports nutrition information that is accurate, complete, clear, reliable, and supported by strong evidence.

## Materials and methods

### Research question

This study aimed to answer the following question: Do AI chatbots provide high-quality sports nutrition information? To answer this question, the accuracy, completeness, clarity, quality of cited evidence, and test-retest reliability of several generative AI chatbots built on different LLMs were determined.

### Study design

The study was planned in August 2024 and the study protocol was registered on the Open Science Foundation (OSF; https://doi.org/10.17605/OSF.IO/ZCKYA) registry on September 9 2024, before data collection began [42]. The study was initiated and data collection commenced in October 2024. Amendments to the original protocol were documented on the OSF registry [42]. The study was designed in line with the METRICS checklist [43] and reported according to METRICS (https://osf.io/59pxz) and the Guidelines for Reporting Reliability and Agreement Studies (GRRAS; https://osf.io/q8n4x) [44].

This paper includes two experiments. ***Experiment 1*** posed two types of single (zero-shot) prompts (*simple* and *detailed*) covering two domains (*Sports nutrition to support training,* and *Sports nutrition for racing*) to six AI chatbots from three chatbot developers. The chatbots' outputs were assessed against sports nutrition guidelines [14–16] to rate *accuracy*, *completeness*, and *additional info quality*. The outputs' *clarity* and *quality of cited evidence* were also assessed, and the *test-retest reliability* of these metrics was determined. In ***Experiment 2***, the same chatbots were provided with a sports nutrition exam and their answers were assessed for *accuracy* and *test-retest reliability*.

### Experiment 1

**Chatbot prompts.**  Table 1 shows the *simple* and *detailed* prompts entered into the chatbots. All prompts were specific to sports nutrition and written in English. The *simple* prompts were single-sentence zero-shot prompts based on phrases searched for on Google Trends (https://trends.google.com/trends/explore); they were designed to reflect the style of conversational prompts entered by nonexpert chatbot users. The detailed prompts were developed using best practice prompting techniques based on prompt engineering principles [28–35]. Consequently, the detailed prompts included a context, a role, a task (including a chain of thought, a list of specifics, and emotional stimuli to emphasise the importance of the prompt), and notes to prevent lost-in-the-middle effects.

Please note that, while the current sports nutrition guidelines mention dietary iron as a nutrient of interest to menstruating females and discuss the concept of energy availability in relation to the Female Athlete Triad and Relative Energy

**Table 1. "Simple" and "detailed" chatbot prompts used in Experiment 1.**

| Type of prompt | Domain | Prompt |
|---|---|---|
| Simple prompts. | A. Sports nutrition to support training. | *What are the sports nutrition guidelines for supporting an endurance athlete's daily training?* |
| | B. Sports nutrition for racing. | *What are the sports nutrition guidelines for improving race performance during a marathon?* |
| Detailed prompts. | A. Sports nutrition to support training. | Context: I'm a highly-trained endurance athlete who competes in marathons. I want to improve my running performance and understand the basic concepts of nutrition but I'm unsure about sports nutrition and how to support my training with ideal nutritional strategies. Role: Your role is a registered sports nutritionist/sports dietician, which is to assess the nutritional needs of athletes and provide dietary prescriptions and education to enhance athletes' health, performance, and body composition. Your role is important and I will greatly value your answer because the recommendations will help improve my success as a marathon runner. Task: In your role, I want you to provide me with sports nutrition recommendations that will help support my training. Specifics: This task is critical to my success as a marathon runner so please provide a thorough answer. In your answer, I want expert-level writing but I don't want you to have opinions. To support your answer, I only want you to use evidence from scientific research papers published in peer-reviewed journals and available on PubMed. I want you to insert numbered references within your answer and provide a list of references that includes DOI numbers, PMIDs, or web links to the papers on PubMed. If you can't provide an answer, tell me that you don't know the answer. Notes: You are a registered sports nutritionist/sports dietician. Please provide sports nutrition recommendations to support my marathon running training. Only use evidence from published scientific research papers. |
| | B. Sports nutrition for racing. | Context: I'm a highly-trained endurance athlete who competes in marathons. I want to improve my marathon running performance and understand the basic concepts of nutrition but I'm unsure about sports nutrition applied to my racing. Role: Your role is a registered sports nutritionist/sports dietician, which is to assess the nutritional needs of athletes and provide dietary prescriptions and education to enhance athletes' health, performance, and body composition. Your role is important and I will greatly value your answer because the recommendations will help improve my success as a marathon runner. Task: In your role, I want you to provide me with specific sports nutrition advice that will help me improve my performance during my next marathon. More specifically, I want you to advise me on sports nutrition best practices for the days before my marathon, the morning of my marathon, and during my marathon. In addition, I want to know how many grams of carbohydrates I should eat on the days before my marathon, the morning of my marathon, and during my marathon. I also want you to provide examples of carbohydrate-containing foods I could eat during my marathon and the morning of my marathon. And, I want you to provide examples of carbohydrate-containing foods or sports nutrition products I could consume the products during the marathon. Specifics: This task is critical to my success as a marathon runner so please provide a thorough answer. In your answer, I want expert-level writing but I don't want you to have opinions. To support your answer, I only want you to use evidence from scientific research papers published in peer-reviewed journals and available on PubMed. I want you to insert numbered references within your answer and provide a list of references that includes DOI numbers, PMIDs, or web links to the papers on PubMed. If you can't provide an answer, tell me that you don't know the answer. Notes: You are a registered sports nutritionist/sports dietician. Please provide specific sports nutrition advice to improve my performance at my next marathon. Include daily grams of carbohydrates and examples of foods for the days before my marathon, the morning of my marathon, and during my marathon. Only use evidence from published scientific research papers. |

Deficiency in Sport (RED-S) [14–16], current sports nutrition position stands and consensus statements do not provide separate guidelines for biological females from biological males. Furthermore, nutritional guidelines given in existing position stands and consensus statements [14–16] already include the sentiments provided in papers describing the needs of female athletes [45]. We acknowledge that this may change given the emerging sex differences in performance nutrition [45]; however, we chose to generalise our prompts and not mention biological sex or gender constructs. Similarly, because sports nutrition guidelines are not race/ethnicity/culture-specific, we chose not to mention such factors in our prompts.

**Prompt inputting.** New accounts were created with publicly available AI chatbots from three companies: OpenAI, Google, and Anthropic. The prompts from Table 1 were entered into these chatbots using their basic and advanced models (Table 2); the models were not fine-tuned. To examine test-retest reliability, the prompts were entered on two occasions, October 7 2024 and October 11 2024, using a Google Chrome browser on a MacOS. The proximity of test-retest dates was chosen to ensure that the same chatbot model was used within manufacturers. There were no model updates between these timeframes [46]. To prevent bias from feedback and learning loop effects, prompts were entered without using chatbot customisations or plug-ins, and the chatbot's history was deleted before each prompt. Where possible, the chatbot's memory was turned off, and a new chat was created each time a new prompt was entered. The output files for each chatbot are available in the data registry at https://osf.io/k2c6t/ [47]. To blind the investigators (TS and ML) to the chatbot identity of the outputs and thereby minimise bias, the prompts were entered by an independent prompter (Lisa Tindle). LT copied each output into a Microsoft Word (.docx) file saved under a coded filename and then sent the.docx files to TS and ML for them to complete the ratings (described below). When creating the.docx files, LT removed any text from the chatbot output that would reveal the chatbot identity to TS and ML.

**Rating chatbot performance.** The following sports nutrition position stands and consensus statements were used to build criteria for rating chatbot performance: (i) the 2016 Nutrition and Athletic Performance Joint Position Statement of the Academy of Nutrition and Dietetics, Dietitians of Canada, and the American College of Sports Medicine (ACSM) [14]. (ii) The 2018 International Society of Sports Nutrition (ISSN) exercise & sports nutrition review update: research & recommendations [15]. And, (iii) the 2019 International Association of Athletics Federations Consensus Statement: Nutrition for Athletics [16]. The set of criteria the authors agreed upon for rating chatbot performance is shown in Table 3.

The expected answers were derived from the 2016 nutrition and athletic performance joint position statement of the Academy of Nutrition and Dietetics, Dietitians of Canada, and the American College of Sports Medicine (ACSM) [14], the 2018 International Society of Sports Nutrition (ISSN) exercise and sports nutrition recommendations [15], and the 2019 International Association of Athletics Federations consensus statement on nutrition for athletics [16].

**Table 2. The chatbots used in this study.**

| Chatbot developer | Basic LLM AI chatbot* | Advanced LLM AI chatbot** |
|---|---|---|
| OpenAI [48] | ChatGPT-4o mini<br>Release date: July 18, 2024 [49].<br>Model name: gpt-4o-mini | ChatGPT-4o<br>Release date: May 13, 2024 [50].<br>Model name: gpt-4o |
| Google [51] | Gemini 1.5 flash<br>Release date: September 24, 2024 [46].<br>Model name: gemini-1.5-flash-002 | Gemini Advanced (1.5 pro)<br>Release date: September 24, 2024 [46].<br>Model name: gemini-1.5-pro-002 |
| Anthropic [52] | Claude 3.5 Sonnet<br>Release date: June 20, 2024 [53].<br>Model name: claude-3–5-sonnet-20240620 | Claude Pro (Claude 3 Haiku)<br>Release date: July 03, 2024 [53].<br>Model name: claude-3-haiku-20240307 |

* The basic AI chatbots from OpenAI, Google, and Anthropic were free of charge at the time of writing.

** The advanced AI chatbots from OpenAI, Google, and Anthropic are subscription-based; at the time of writing, each one cost USD $20 per month.

**Table 3. Sports nutrition criteria for rating chatbot performance in Experiment 1.**

| Domain | Criteria and corresponding expected answers |
|---|---|
| A. Sports nutrition to support training. | 1. *Daily energy availability*. A statement advising that athletes should consume a daily diet that is adequate in energy availability and nutrient provision to prevent low energy availability.<br>2. *Daily carbohydrate intake*. A statement suggesting a daily carbohydrate intake in the range of 3–10 grams per kilogram body weight per day (g/kg/day), and up to 12 g/kg/day for extreme and prolonged activities.<br>3. *Daily protein intake*. A statement suggesting a daily protein intake in the range of 1.2 to 2.0 g/kg/day, or a statement suggesting a protein intake of approximately 0.3 g/kg every 3–4 h across the day.<br>4. *Post-session carbohydrate intake*. A statement advising to start restoring muscle glycogen soon after demanding sessions with foods/fluids that provide carbohydrates at approximately 1.0 to 1.2 g/kg per hour for 4–6 hours after the session.<br>5. *Post-session protein intake*. A statement advising to consume foods/fluids soon after sessions that provide 0.25 to 0.55 g/kg (or 20–40 g) of protein.<br>6. *Hydration*. A statement advising to consume fluids during the day to quench thirst and prevent dehydration, and to consume fluids after sessions to replenish fluid levels and maintain hydration.<br>7. *Supplements*. A statement suggesting that a daily multivitamin/multimineral supplement *might* be necessary if Recommended Dietary Allowance (or Adequate Intake) values are not met by daily nutrient intake, and a statement suggesting that a specific vitamin/mineral supplement (iron, B12, etc.) might also be necessary if the athlete has a nutritional deficiency, special requirement, or follows a plant-based diet.<br>8. *Individualization*. A statement advising that nutrition should be individualized/personalized according to the athlete's training demands and event (i.e., *fuel for the work required*, *fuel for the intended adaptation*, *do what works for you*, etc.).<br>9. *Disclaimer*. A statement advising to consult with a registered dietitian or sports nutritionist for individualized/personalized advice. |
| B. Sports nutrition for racing. | 1. *Carbohydrate intake on the days before the race*. A statement advising to consume a carbohydrate-rich diet providing 7–12 g/kg/day of carbohydrates for 24–48 hours before the race (to increase muscle glycogen).<br>2. *Examples of carbohydrate-containing foods for the days before the race*. A statement suggesting to consume cereals, pasta, rice, bread, potatoes, polenta, couscous, fruit, etc.<br>3. *Carbohydrate intake during the hours before the race*. A statement advising to consume carbohydrate-containing foods/fluids providing 1–4 g/kg of carbohydrates in the 1–4 hours before the race (to restore liver glycogen after an overnight fast).<br>4. *Examples of carbohydrate-containing foods for the hours before the race*. A statement suggesting consuming low-fibre, easily digestible foods like cereals, bread, toast, bagels, fruit (bananas), honey, syrups, sports drinks, etc.<br>5. *Carbohydrate intake during the race*. A statement advising to regularly consume carbohydrate-containing foods/fluids that provide carbohydrates at an intake of 30–90 g/hour during the race (to maintain high carbohydrate availability).<br>6. *Examples of carbohydrate-containing foods and sports nutrition products for during the race*. A statement suggesting to consume sports drinks, gels, chews, bars, candy, fruit, etc.<br>7. *Hydration before the race*. A statement advising to consume fluids during the 1–4 hours before the race (to ensure adequate hydration).<br>8. *Hydration during the race*. A statement advising the athlete to develop an individualized fluid plan that uses the opportunities to drink during a race with the goal to replace as much sweat loss as is practical while avoiding drinking fluids in excess of sweat rate.<br>9. *Supplements*. A statement suggesting to consider supplementation with caffeine (pre- or during-race) and/or nitrate (pre-race) to help improve endurance performance.<br>10. *Individualization*. A statement advising that the race day nutrition plan should be well-practised and individualized according to the athlete's preferences, tolerance, and experiences (i.e., *do what works for you*).<br>11. *Disclaimer*. A statement advising to consult with a registered dietitian or sports nutritionist for individualized/personalized advice. |

For both types of prompts (*Simple* and *Detailed*) and for both domains (*Sports nutrition to support training* and *Sports nutrition for racing*), the outputs from each chatbot from the test and retest were independently rated, subjectively, by both authors (TS and ML) using Likert scales in Google Forms. The raters have over 15 years of experience in nutritional science and sports nutrition. *Accuracy*, *completeness*, *clarity*, *additional info quality*, and *quality of cited evidence* were rated as follows. Similar procedures have been used previously [17,25–27].

The *accuracy* of the response outputs was assessed with a 5-point Likert scale to rate each criterion in Table 3, where:

1 = Answer includes *no* aspects of the expected answer. (low accuracy)

2 = Answer includes *some* (less than half) aspects of the expected answer (more incorrect than correct).

3 = Answer includes *about half* of the expected answer (approximately equally correct and incorrect; moderate accuracy).

4 = Answer includes *most* (more than half) aspects of the expected answer (more correct than incorrect).

5 = Answer includes *all* aspects of the expected answer. (high accuracy)

The *completeness* of the response outputs was assessed by rating the depth of each chatbot's output using a 3-point Likert scale, where:

1 = Inadequate and incomplete, addresses some aspects of the question, but significant parts are missing or incomplete. (low completeness)

2 = Adequate and complete, addresses all aspects of the question and provides the minimum amount of information required to be considered complete. (moderate completeness)

3 = More than adequate, comprehensively addresses all aspects of the question <u>and</u> provides additional information or context beyond the expected answer in Table 3. (high completeness)

If the information provided in the chatbot's output was beyond the scope of the criteria in Table 3, additional information quality was rated against consensus evidence in the position stands and consensus statements from ACSM, ISSN, and IAAF [14–16]. This was included to help capture potentially erroneous and, therefore, misleading information provided by chatbots. A 5-point Likert scale was used to rate *additional information quality*, where:

1 = Additional information is completely erroneous, not at all supported by consensus evidence (from ACSM, ISSN, and IAAF) and completely misleading. (low additional information quality)

2 = Additional information is somewhat erroneous, mostly unsupported by consensus evidence, and largely misleading.

3 = Additional information is mixed: about half of the additional information is (somewhat) accurate and (partially) supported by consensus evidence while half of the additional info is inaccurate and unsupported by consensus evidence. (moderate additional information quality)

4 = Additional information is somewhat accurate, partially supported by consensus evidence, and potentially useful.

5 = Additional information is accurate, supported by consensus evidence (from ACSM, ISSN, and IAAF), and useful. (high additional information quality)

The *clarity and coherence* of the outputs were assessed by rating how logical, clear, and understandable they were. A 3-point Likert scale was used, where:

1 = Answer is disorganised, difficult to understand, and poorly written with many typographical/grammatical errors. (low clarity and coherence)

2 = Answer is somewhat well-organised, mostly easy to understand, and generally well-written but includes some typographical/grammatical errors. (moderate clarity and coherence)

3 = Answer is very well-organised, very easy to understand, and very well-written with no typographical/grammatical errors. (high clarity and coherence)

The *quality of cited evidence* was assessed by rating the citations provided in the outputs using a 3-point Likert scale, where:

1 = Answer includes either *no* citations or, if citations are included, they are *mostly* incorrect (irrelevant to the content) and/or hallucinations. (low quality of cited evidence)

2 = Answer includes *some* citations but either there is no reference list after the answer, or *some* (less than half) of the references are incorrect/hallucinations, or the references are not papers published in peer-reviewed journals, or the reference list does not include DOI numbers, PMIDs, or web links to the papers. (moderate quality of cited evidence)

3 = Answer includes citations with numbered references to scientific research papers published in peer-reviewed journals, <u>and</u> the references are real (not hallucinated/fabricated) and correct (relevant to the content), <u>and</u> there is a list of references after the answer that includes DOI numbers, PMIDs, or web links to the papers on PubMed or the journal's webpage. (high quality of cited evidence)

The Google Forms ratings were automatically stored in a Google Sheets file and author TS saved it as a CSV file for statistical analysis (available in the data registry at https://osf.io/k2c6t/ [47]). After output ratings and statistical analyses were complete, LT broke the code to unblind the data and reveal the identity of the chatbots.

The primary outcome was between-chatbot overall accuracy. The secondary outcomes were domain-specific accuracy, completeness, clarity, additional info quality, evidence quality, and the between-chatbot test-retest reliability of all the above-listed metrics.

## Experiment 2

**Chatbot prompts.** An exam containing a set of n = 111 multiple-choice questions (MCQ) with a single correct answer was entered as a prompt into the same chatbots listed in Table 2. As for Experiment 1, the models were not fine-tuned. All questions were specific to sports nutrition and written in English. The questions were obtained from My Sports Dietitian [54] and resembled those asked on the *Certified Specialist in Sports Dietetics* (CSSD) board exam. The questions cover 3 domains: *Exercise and performance nutrition*, *Clinical sports nutrition*, and *Nutrition operation and management*.

**Prompt inputting.** The prompts were entered into the chatbots shown in Table 2 using a Google Chrome browser on a MacOS. The chatbot prompt included the text *"Please answer the following multiple-choice questions by selecting the answer you think is correct. Please output your answers in a CSV file format with the question numbers in column A and your answer in column B."*, followed by the list of MCQ exam questions. To examine test-retest reliability, the prompts were entered on October 7 2024 and October 11 2024. The proximity of test-retest dates ensured that the same chatbot model was used within manufacturers; indeed, there were no model updates between these timeframes [46]. The order of the MCQs for the first test trial was randomised for each chatbot developer but identical between the basic and advanced models. For example, the MCQ order was identical between ChatGPT-4o mini (basic OpenAI model) and ChatGPT-4o (advanced OpenAI model) but different between models from OpenAI, Google, and Anthropic. The order of the MCQs used in the first test trial was re-used in the retest trial. MCQ order randomisation was achieved using the RAND() function in Google Sheets.

To prevent bias from feedback and learning loop effects, prompts were entered without using chatbot customisations or plug-ins, and the chatbot's history was deleted before each prompt. Where possible, the chatbot's memory was also turned off and a new chat was created each time a new prompt was entered. Unfortunately, the exam questions cannot be shared, in line with our non-disclosure agreement with My Sports Dietitian. To blind the investigators (TS and ML) to the chatbot identity of the outputs and thereby minimise bias, the prompts were entered by LT who saved the output files (.txt) under a coded filename. The output files for each chatbot are available in the data registry at https://osf.io/k2c6t/ [47]. When creating the files, LT removed any text from the chatbot output that would reveal the chatbot identity to TS and ML.

**Rating chatbot performance.** LT sent the output TXT files from each chatbot on each occasion to TS who merged them into a single master CSV file using R. The MCQ answers in the master file were rated objectively using an automated script to minimise bias, where a correct answer = 1 point and an incorrect answer = zero points. The primary outcome was between-chatbot exam accuracy (the proportion of correct exam answers). The secondary outcomes were domain-specific accuracy and the between-chatbot test-retest reliability of exam accuracy.

## Statistical analyses

Power calculations were made using G*Power v3.1.9.7 [55]. To detect a large effect for 2 repeated measures between 6 chatbots with alpha = 0.05 and 1-beta (power) = 0.80, a total sample of 90 was required. In Experiment 1, there were 20 observations (criteria/expected answers) so the total sample size = 120 (6 chatbots × 20), which is adequately powered to detect large between-chatbot effects and will likely detect moderate effects. In Experiment 2, the exam contained 111 observations (exam questions), which means that our total sample size of 666 (6 chatbots × 111) was adequately powered to detect large between-chatbot effects and will likely detect small effects.

All statistical analyses were performed by author TS using R version 4.4.2 [56] on Google Colab. The following R packages were used: *stats* (part of R version 4.4.2) *tidyverse* [57], *broom* [58], *infer* [59], *ggpubr* [60], *irr* [61], *lme* [62], *lmerTest* [63], *lmtest* [64], *car* [65], *effectsize* [66], *emmeans* [67], and *pwr* [68].

**Statistical analyses for Experiment 1.** Because the Likert scale was ordinal, Likert values for individual criteria were reported as the median ± interquartile range (IQR) of the raters' scores. However, when compiling lists of scores, the Likert scale is assumed to become continuous and, therefore, the mean of individual criteria Likert scores was presented as mean ± standard deviation [69]. An intraclass correlation coefficient (ICC) with a 95% confidence interval (CI) was calculated using a *two-way mixed effects* model with *absolute agreement* as the relationship among raters and *mean of raters* as the unit of interest. The ICC was interpreted to indicate poor (ICC < 0.50), moderate (0.5 ≤ ICC < 0.75), good (0.75 ≤ ICC < 0.9), or excellent (ICC ≥ 0.9) reliability [70].

For each type of prompt (*Simple* and *Detailed*), domain-specific (*Training* and *Racing*) and overall accuracy scores were calculated for each LLM on each occasion. The accuracy scores were calculated as the mean ± standard deviation of the Likert scores from each criterion in the *Sports nutrition for training* domain (9 criteria), *Sports nutrition for racing* domain (11 criteria), and overall (*training + racing* domains; 20 criteria). Percentage accuracy scores were then calculated and interpreted in line with the original Likert scale, as follows: < 20% = Answer includes *no* aspects of the expected answer (low accuracy); > 20% to ≤ 40% = Answer includes *some* (less than half) aspects of the expected answer (more incorrect than correct); > 40% to ≤ 60% = Answer includes *about half* of the expected answer (approximately equally correct and incorrect; moderate accuracy); > 60% to ≤ 80% = Answer includes *most* (more than half) aspects of the expected answer (more correct than incorrect); > 80% = Answer includes *all* aspects of the expected answer. (high accuracy).

For each type of prompt (*Simple* and *Detailed*), domain-specific (Training and Racing) and overall scores for completeness, clarity, additional info quality, and quality of cited evidence were reported for each LLM on each occasion. Note that these 4 scores were assessed for the entire chatbot output, not for each criterion in Table 3 as was done for accuracy scores. Because these 4 scores are single observations, no statistical models were used to analyse them. On the contrary, accuracy scores were analysed as follows:

Shapiro–Wilk and Bartlett tests were used to determine the normality and homogeneity of variance of the accuracy scores, which were log-transformed if the data deviated from parametric assumptions. Mixed-methods repeated-measures ANOVAs (linear mixed models) were used to compare *Accuracy* scores within (test-retest reliability) and between chatbots, with *Domain, PromptType, TestDay,* and their interactions set as fixed effects, and *ChatbotID* and random slopes for *Test-Day* set as random effects. Bonferroni-adjusted pairwise t-tests were used to control the family-wise error rate for multiple comparisons. Due to the complexity of the data set, a chi-squared log-likelihood test was used to compare the goodness of fit of linear mixed models with and without random slopes to a four-way ANOVA (*ChatbotID × Domain × PromptType × TestDay*) with Bonferroni-adjusted *posthoc* comparisons. The Akaike Information Criterion (AIC) and the Bayesian Information Criterion (BIC) were calculated to estimate the prediction error of the models. The model with the greatest log-likelihood and lowest AIC and BIC values was selected to interpret the data.

**Statistical analyses for Experiment 2.** First, the Kuder-Richardson 20 (KR-20) statistic was calculated to measure the internal consistency reliability of the exam [71]. KR-20 was interpreted as follows: KR-20 ≥ 0.90 = excellent reliability, 0.80 ≤ KR-20 < 0.90 = good reliability, 0.70 ≤ KR-20 < 0.80 = acceptable reliability, 0.60 ≤ KR-20 < 0.70) = questionable

reliability, $0.50 \leq \text{KR-20} < 0.60 = \text{poor reliability}$, $\text{KR-20} < 0.50 = \text{unacceptable reliability}$. Next, an overall accuracy score was calculated for each LLM on each occasion as the proportion of correct answers to the 111 exam questions. Therefore, the accuracy score was a continuous value between 0 and 1, which is equal to a percentage accuracy mark when multiplied by 100. Because the pass/fail mark for the CSSD board exam varies for each version of the exam [72,73] it was not possible to assign a percentage score that demarcates a binary pass/fail result. Domain-specific accuracy scores were also calculated for each of the 3 domains of the exam: *Exercise and performance nutrition* (Domain A, 66 questions), *Clinical sports nutrition* (Domain B, 38 questions), and *Nutrition operation and management* (Domain C, 7 questions).

A mixed-methods repeated-measures ANOVA (a logistic mixed model with a binomial distribution for the binary outcome, exam answer score: 1=correct, 0=incorrect) was then used to compare exam answer scores within (test-retest reliability) and between chatbots, with *ChatbotID*, *ExamDomain*, *TestDay*, and their interactions set as fixed effects and *ChatbotID* set as a random intercept for each chatbot to account for repeated measures. A similar logistic mixed model with random slopes added for *TestDay* was also examined. A Generalized Linear Model (GLM) with a binomial distribution (a standard logistic regression model without random effects) was also used to examine the fixed effects of *ChatbotID, ExamDomain,* and *TestDay, and their interactions*. Bonferroni-adjusted pairwise t-tests were used to control the family-wise error rate for multiple comparisons. A chi-squared log-likelihood test was used to compare the goodness of fit of the models and the AIC and BIC were calculated to estimate the prediction error of the models. The model with the best fit and lowest AIC and BIC values was chosen to interpret the data.

Statistical significance was achieved when $P \leq 0.05$. Effect sizes with 95% confidence intervals and statistical power for alpha=0.05 were also calculated. Effect sizes were interpreted to infer trivial ($\eta^2 < 0.01$; Cohen's $d < 0.20$; $r < 0.10$), small ($0.01 \leq \eta < 0.06$; $0.20 \leq d < 0.50$; $0.01 \leq r < 0.30$), moderate ($0.06 \leq \eta^2 < 0.14$; $0.50 \leq d < 0.80$; $0.30 \leq r < 0.50$), or large effects ($\eta^2 > 0.14$; $d \geq 0.80$; $r > 0.50$).

## Modifications to the original protocol

It was originally planned to include Microsoft Copilot in this study. However, the character limit of the prompt box for Microsoft Copilot prevented the input of the detailed prompts in Experiment 1 and the exam questions in Experiment 2. Furthermore, Copilot's basic model did not have a PDF upload function. For these reasons, we chose to remove Copilot from the study, as documented on the OSF registry [42].

## Results

### Experiment 1

Agreement between the two raters was good (ICC=0.893, 95% CI 0.869 to 0.912; F=9.82, p<0.001) so the mean of the rater's scores was calculated and reported. However, due to non-normally distributed data and unequal variances across groups, data were log-transformed before statistical analyses. An ANOVA model was chosen to interpret the accuracy scores in Experiment 1 because it had a better fit (highest log likelihood) and lower prediction errors (smaller AIC and BIC values) compared to the linear mixed model. There was a statistically significant main effect of ChatbotID (F=7.66, p=6.52 × 10−7, partial η2=0.0814, 95% CI 0.0376 to 1.00) but no statistically significant main effects for Domain (p=0.671), PromptType (p=0.723), TestDay (p=0.917), or interactions among them (p=0. 0651 to 1.00). Therefore, differences in accuracy scores existed between chatbots, but accuracy scores were not influenced by the domain (*Sports nutrition for Training* vs. *Sports nutrition for racing*) or the type of prompt (*Simple* vs. *Detailed*). Furthermore, the lack of difference in accuracy scores between the two test days (days 1 and 2) demonstrates good rest-retest reliability. See S1 Table in S1 File for the ANOVA summary table and S2 Table in S1 File for the ANOVA model's partial eta-squared values (all supplemental tables available at https://osf.io/qp98k [47]).

With a simple prompt, Claude3.5Sonnet and ClaudePro showed low accuracy (less than half of the aspects of the expected answer), but this was improved to moderate accuracy with a detailed prompt (Fig 1). All other chatbots showed moderate accuracy for both simple and detailed prompts, with their outputs including more than half of the expected

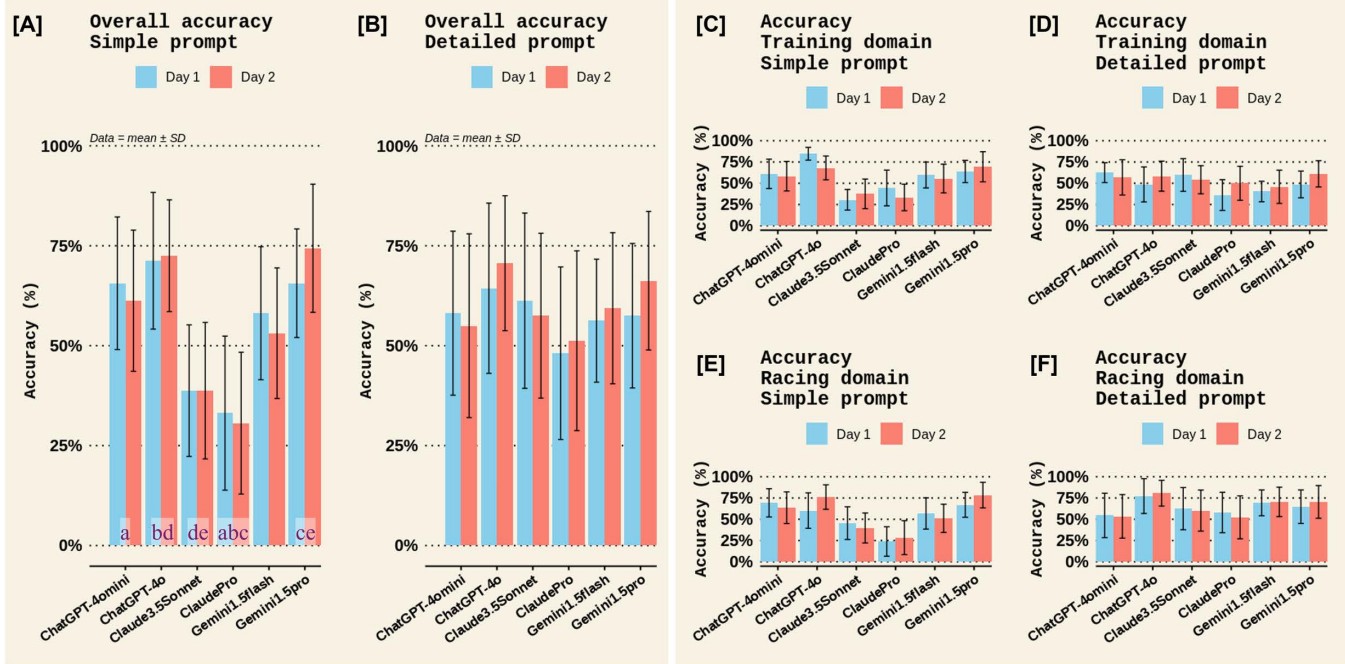

**Fig 1. Accuracy scores among different chatbots on the two test days in Experiment 1.** Panels [A] and [B] display the overall accuracy scores across both domains (*Sports Nutrition for Training* and *Sports Nutrition for Racing*) for the *Simple* and *Detailed* prompts, respectively. Panels [C] and [D] display the accuracy scores in the *Training* domain for the *Simple* and *Detailed* prompts, respectively, while panels [E] and [F] show the accuracy scores in the *Racing* domain. Bars represent the mean of accuracy scores for each criterion and error bars represent the standard deviation (SD). ANOVA revealed a significant main effect of Chatbot ID where accuracy scores for ChatGPT-4omini (comparison "a": p=0.008, d=0.549), ChatGPT-4o ("b": p<0.001, d=0.796), and Gemini1.5pro ("c": p<0.001, d=0.752) were greater than ClaudePro, and the accuracy scores for ChatGPT-4o ("d": p=0.008, d=0.546) and Gemini1.5pro ("e": p=0.02, d=0.502) were greater than Claude3.5Sonnet.

answers; however, none of the chatbots had a high level of accuracy (Fig 1). Post hoc comparisons on ChatbotID, independent of the type of prompt, found that the accuracy scores for ChatGPT-4omini (p=0.008, d=0.549), ChatGPT-4o (p<0.001, d=0.796), and Gemini1.5pro (p<0.001, d=0.752) were greater than for ClaudePro but not different from one another (Tables 4, 5, and Fig 1). Furthermore, ChatGPT-4o (p=0.008, d=0.546) and Gemini1.5pro (p=0.02, d=0.502) had greater accuracy scores than Claude3.5Sonnet (Tables 4, 5, and Fig 1). The comparisons showed moderate effect sizes, but statistical power to detect significant differences was only low to moderate (Table 5). Full data for the post hoc comparisons of ChatbotID, TestDay, Domain, PromptType, and the ChatbotID×PromptType interaction are available in S3–S6 Tables in S1 File. Further post hoc comparisons on the ChatbotID×PromptType interaction (p=0.0651, partial η2=0.0237) revealed that the above-described between-chatbot differences in accuracy were only evident within the simple prompts (Gemini1.5flash vs. ClaudePro: p=0.0507, d=0.733; ChatGPT-4o vs. ClaudePro: p=0.0001, d=1.09; ChatGPT-4omini vs. ClaudePro: p=0.005, d=0.889; Gemini1.5pro vs. ClaudePro: p=0.0002, d=1.07; ChatGPT-4o vs. Claude3.5Sonnet: p=0.0087, d=0.856; Gemini1.5pro vs. Claude3.5Sonnet: p=0.0123, d=0.835), not the detailed prompts. This suggests the between-chatbot differences observed with simple prompts were not evident when detailed prompts were used. Full post hoc data for the ChatbotID×PromptType interaction is available in S7 Table in S1 File.

A qualitative examination of the criterion used to evaluate accuracy was made. For the *Sports nutrition for Training* domain, most chatbots scored highly for daily carbohydrate intake, daily protein intake, and individualisation; moderately for post-session carb intake, post-session protein intake, and hydration; and, poorly for information on daily energy availability and supplements, with highly variable scores for including a disclaimer to seek advice from a registered dietician

**Table 4. Chatbot accuracy scores in Experiment 1.**

| Chatbot | Test day | Accuracy scores Simple prompt | | Accuracy scores Detailed prompt | | Contrasts |
|---|---|---|---|---|---|---|
| | | mean | SD | mean | SD | |
| ChatGPT-4omini | Day1 | 66% | 17% | 58% | 21% | a |
| ChatGPT-4omini | Day2 | 61% | 18% | 55% | 23% | |
| ChatGPT-4o | Day1 | 71% | 17% | 64% | 21% | bd |
| ChatGPT-4o | Day2 | 73% | 14% | 71% | 17% | |
| Claude3.5Sonnet | Day1 | 39% | 16% | 61% | 22% | de |
| Claude3.5Sonnet | Day2 | 39% | 17% | 58% | 21% | |
| ClaudePro | Day1 | 33% | 19% | 48% | 22% | abc |
| ClaudePro | Day2 | 31% | 18% | 51% | 22% | |
| Gemini1.5flash | Day1 | 58% | 17% | 56% | 15% | |
| Gemini1.5flash | Day2 | 53% | 16% | 59% | 19% | |
| Gemini1.5pro | Day1 | 66% | 14% | 58% | 18% | ce |
| Gemini1.5pro | Day2 | 74% | 16% | 66% | 17% | |

SD = standard deviation.

Contrasts: ChatGPT-4omini (comparison "a": p = 0.008, d = 0.549), ChatGPT-4o ("b": p < 0.001, d = 0.796), and Gemini1.5pro ("c": p < 0.001, d = 0.752) were greater than ClaudePro. ChatGPT-4o ("d": p = 0.008, d = 0.546) and Gemini1.5pro ("e": p = 0.02, d = 0.502) were greater than Claude3.5Sonnet.

**Table 5. Contrasts for ChatbotID in Experiment 1.**

| Contrast | P-value | | Cohen's D | 95%CI | | Power |
|---|---|---|---|---|---|---|
| ChatGPT-4o – ClaudePro | <.0001 | *** | 0.796 | 0.479 | 1.113 | 0.689 |
| ClaudePro – Gemini1.5pro | <.0001 | *** | −0.752 | −1.068 | −0.435 | 0.639 |
| ClaudePro – ChatGPT-4omini | 0.008 | ** | −0.549 | −0.864 | −0.235 | 0.395 |
| ChatGPT-4o – Claude3.5Sonnet | 0.008 | ** | 0.546 | 0.232 | 0.861 | 0.392 |
| Gemini1.5flash – ClaudePro | 0.011 | * | 0.532 | 0.218 | 0.847 | 0.375 |
| Gemini1.5pro – Claude3.5Sonnet | 0.021 | * | 0.502 | 0.188 | 0.816 | 0.340 |
| ChatGPT-4omini – Claude3.5Sonnet | 0.414 | | 0.299 | −0.014 | 0.612 | 0.152 |
| Gemini1.5flash – Claude3.5Sonnet | 0.483 | | 0.282 | −0.031 | 0.595 | 0.140 |
| Gemini1.5flash – ChatGPT-4o | 0.558 | | −0.264 | −0.577 | 0.049 | 0.129 |
| ClaudePro – Claude3.5Sonnet | 0.617 | | −0.250 | −0.563 | 0.063 | 0.120 |
| ChatGPT-4o – ChatGPT-4omini | 0.628 | | 0.247 | −0.066 | 0.560 | 0.119 |
| Gemini1.5flash – Gemini1.5pro | 0.738 | | −0.220 | −0.532 | 0.093 | 0.104 |
| ChatGPT-4omini – Gemini1.5pro | 0.798 | | −0.203 | −0.515 | 0.110 | 0.096 |
| ChatGPT-4o – Gemini1.5pro | 1.000 | | 0.045 | −0.268 | 0.357 | 0.052 |
| Gemini1.5flash – ChatGPT-4omini | 1.000 | | −0.017 | −0.329 | 0.295 | 0.050 |

CI = confidence interval of Cohen's D; *, **, and *** indicate significance at 5%, 1%, and 0.1% levels.

(Figs 2 and 3). More specifically, ChatGPT-4omini, ClaudeSonnet, ClaudePro, and Gemini1.5Flash scored poorly on information about energy availability (Fig 2A, 2B), ClaudePro scored poorly on information about supplements (Fig 3A, 3B), and ClaudeSonnet and ClaudePro scored poorly on both individualisation and disclaimer information (Fig 3C–3F). Meanwhile, in the *Sports nutrition for Racing* domain, most chatbots scored highly for carb intake on the days before the race, carb intake in the hours before the race, carb intake during the race, and suggestions for foods, particularly with a detailed prompt (Figs 4 and 5). Most chatbots scored highly for information on individualisation (with notable exceptions

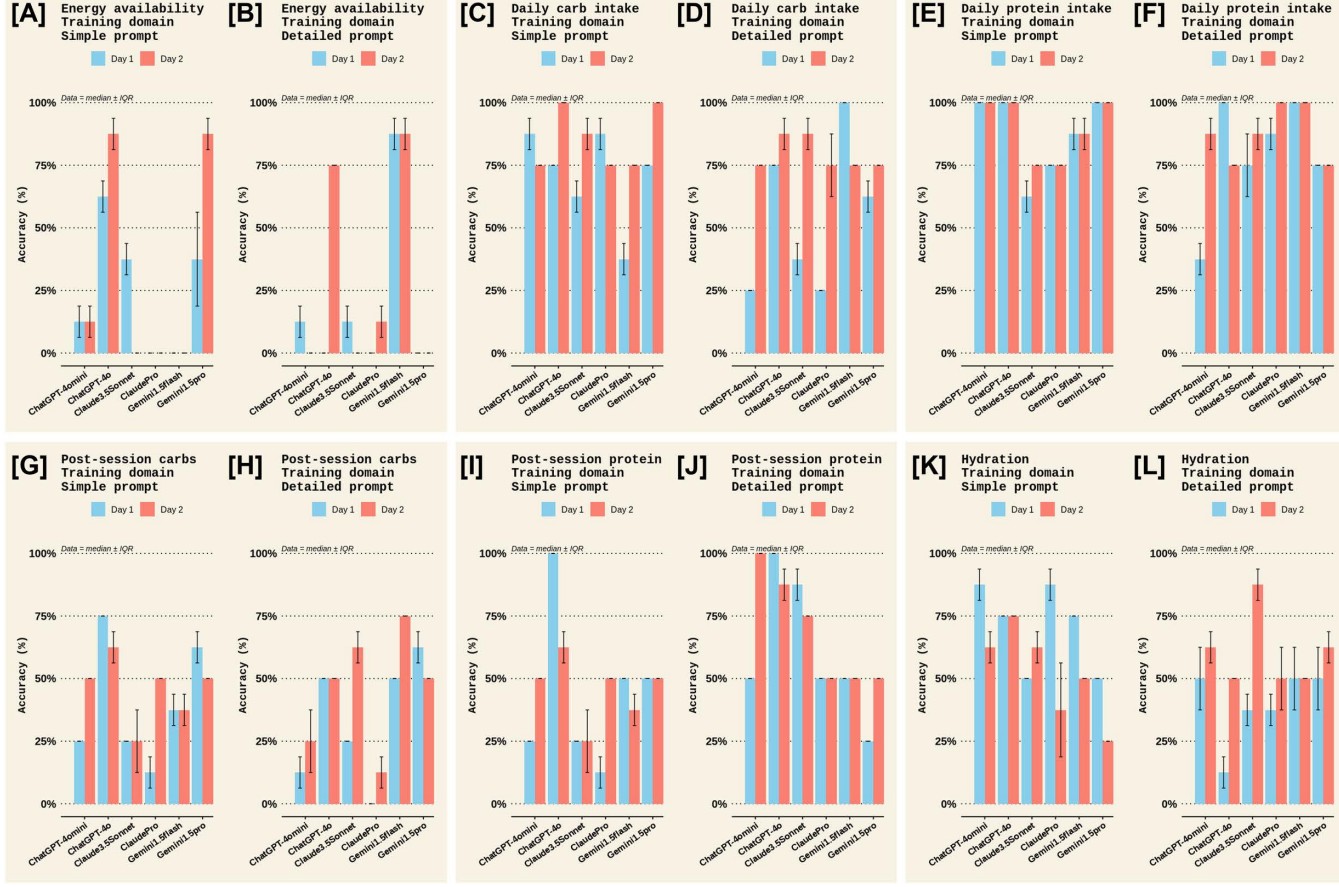

**Fig 2. Accuracy criteria in the *Training* domain among different chatbots on the two test days in Experiment 1.** The accuracy scores for all criteria measured in the *Sports Nutrition for Training* domain. Panels [A] and [B] show accuracy for the *Energy availability* criterion for the *Simple* and *Detailed* prompts, respectively. Panels [C] and [D] show accuracy for the *Daily carbohydrate intake* criterion for the *Simple* and *Detailed* prompts, respectively. Panels [E] and [F] show accuracy for the *Daily protein intake* criterion for the *Simple* and *Detailed* prompts, respectively. Panels [G] and [H] show accuracy for the *Post-session carbohydrate intake* criterion for the *Simple* and *Detailed* prompts, respectively. Panels [I] and [J] show accuracy for the *Post-session protein intake* criterion for the *Simple* and *Detailed* prompts, respectively. Panels [K] and [L] show accuracy for the *Hydration* criterion for the *Simple* and *Detailed* prompts, respectively. Each criteria score is the median ± interquartile range (IQR) of the two raters' scores; therefore, no statistical analyses could be made.

for ClaudePro and ChatGPT-4omini, Fig 5G, 5H) but scored poorly to moderately for information on hydration, and poorly for information on supplements and disclaimers (Figs 4 and 5). Specifically, ClaudeSonnet, ClaudePro, and Gemini1.5flash scored poorly on daily food examples (Fig 4C), ClaudeSonnet and ClaudePro scored poorly on pre-race food examples (Fig 4G), ChatGPT-4omini, ClaudeSonnet, and ClaudePro scored poorly on hydration (Fig 5A–5D), all chatbots except ChatGPT-4o scored poorly on supplements (Fig 5E, 5F), and all chatbots except Gemini1.5Flash and Gemini1. 5Pro scored poorly on disclaimers (Fig 5I, 5J).

A qualitative examination was also made for the remaining outcomes in Experiment 1. For Completeness, most chatbots scored low to moderate completeness scores, except for ChatGPT4o, which scored a moderate to high completeness score (Fig 6). There were no notable differences between prompt types or test days for completeness (Fig 6). All chatbots scored moderate to high clarity scores, with no notable difference between chatbots, prompt types, or test days. (Fig 7). The quality of cited evidence was rated as low for all chatbots when a simple prompt was posed but moderate

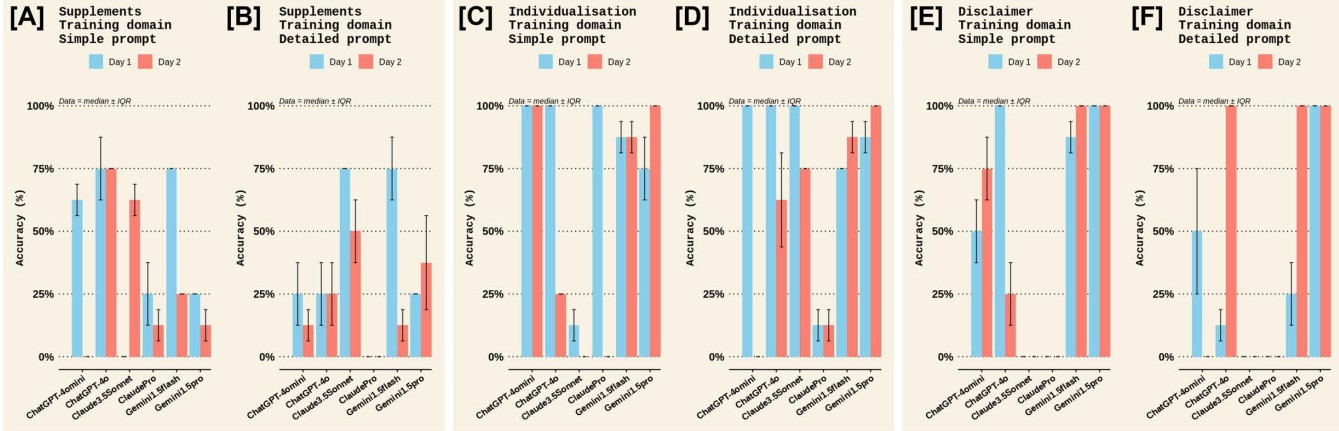

**Fig 3. Accuracy criteria in the *Training* domain among different chatbots on the two test days in Experiment 1 (continued).** The accuracy scores for all criteria measured in the *Sports Nutrition for Training* domain. Panels [A] and [B] show accuracy for the *Supplements* criterion for the *Simple* and *Detailed* prompts, respectively. Panels [C] and [D] show accuracy for the *Individualisation* criterion for the *Simple* and *Detailed* prompts, respectively. Panels [E] and [F] show accuracy for the *Disclaimer* criterion for the *Simple* and *Detailed* prompts, respectively. Each criteria score is the median ± inter-quartile range (IQR) of the two raters' scores; therefore, no statistical analyses could be made.

to high when a detailed prompt was used, with the Claude chatbots outperforming other chatbots (Fig 8). There were no notable differences between test days for the quality of cited evidence. (Fig 8). Lastly, the quality of additional information was moderate for all chatbots in the *Sports nutrition for Training* domain; however, Claude chatbots scored low for this criterion in the *Sports nutrition for Racing* domain (Fig 9). There were no differences between prompt types or test days for additional info quality (Fig 9).

## Experiment 2

The KR-20 value was 0.895, indicating good internal reliability of the exam. This suggests that the exam questions were measuring the same underlying construct consistently; therefore, the exam was effectively assessing the knowledge it was designed to measure. A logistic mixed model without random slopes was chosen to interpret the exam results in Experiment 2 because it had the lowest AIC and BIC prediction errors compared to a logistic model with random slopes and a generalized linear model with a binomial distribution, neither of which improved the log-likelihood of the model fit. This model also met the assumptions for a binomial model: the uniformity of residuals was confirmed via visual inspection of QQ plots and histograms, and model diagnostics tests confirmed that residual distribution was uniform and lacked overdispersion without zero inflation.

In the logistic mixed model, the random intercept for ChatbotID was zero, indicating negligible variation in baseline performance among the individual chatbots beyond that captured by the fixed effects (see below). Therefore, additional modelling of chatbot-level random variability was unnecessary, and findings regarding chatbot performance are generalizable across individual chatbots rather than influenced by idiosyncratic chatbot-specific factors.

For the fixed effects, the intercept was positive and statistically significant (estimate = 1.34, p < 0.001), suggesting a high baseline probability of correct answers (see S8 Table in S1 File for the model summary). Pairwise comparisons showed that Claude3.5Sonnet scored the highest proportion of correct answers to the exam questions (Table 6, Fig 10), outperforming ClaudePro (p < .0001), Gemini1.5flash (p < .0001), ChatGPT-4omini (p = 0.0001), and ChatGPT-4o (p = 0.04; Tables 6, 7, and Fig 10). The second highest scoring chatbot, Gemini1.5pro, also outperformed ClaudePro (p = 0.0001), Gemini1.5flash (p = 0.002), and ChatGPT-4omini (p = 0.004; Tables 6, 7, and Fig 10). These comparisons all had large effect

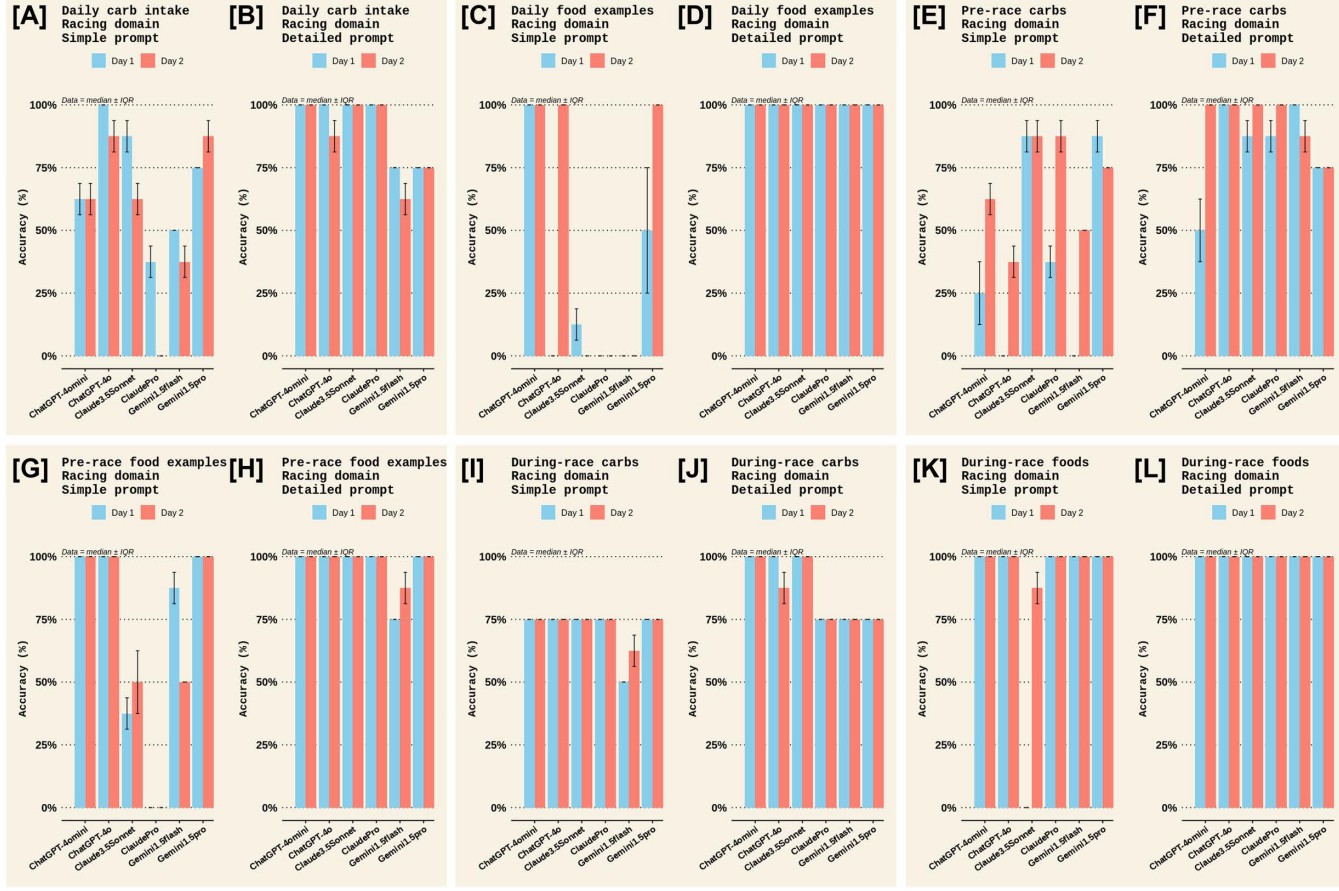

**Fig 4. Accuracy criteria in the *Racing* domain among different chatbots on the two test days in Experiment 1.** The accuracy scores for all criteria measured in the *Sports Nutrition for Racing* domain. Panels [A] and [B] show accuracy for the *Daily carbohydrate intake* criterion for the *Simple* and *Detailed* prompts, respectively. Panels [C] and [D] show accuracy for the *Daily food examples* criterion for the *Simple* and *Detailed* prompts, respectively. Panels [E] and [F] show accuracy for the *Pre-race carbohydrate intake* criterion for the *Simple* and *Detailed* prompts, respectively. Panels [G] and [H] show accuracy for the *Pre-race food examples* criterion for the *Simple* and *Detailed* prompts, respectively. Panels [I] and [J] show accuracy for the *During-race carbohydrate intake* criterion for the *Simple* and *Detailed* prompts, respectively. Panels [K] and [L] show accuracy for the *During-race food examples* criterion for the *Simple* and *Detailed* prompts, respectively. Each criteria score is the median ± interquartile range (IQR) of the two raters' scores; therefore, no statistical analyses could be made.

sizes and high statistical power (Table 5). Furthermore, the proportion of correct answers to the exam questions was lower in the *Clinical sports nutrition* domain compared to the *Exercise and performance nutrition* domain (p < 0.0001) and the *Nutrition operation and management* domain (p = 0.0005; Table 8). However, there was no significant difference between the two test days (p = 0.539; Table 9), demonstrating good rest-retest reliability. See S9–S14 Tables in S1 File for full data.

## Discussion

This study used two experiments to determine the accuracy (validity) and test-retest reliability of several publicly accessible AI chatbots. In *Experiment 1*, chatbots from OpenAI (ChatGPT4omini and ChatGPT4o) and Google (Gemini1.5flash and Gemini1.5Pro) had better accuracy than the chatbots from Anthropic (Claude3.5Sonnet and ClaudePro) when posed with simple zero-shot prompts, but a detailed zero-shot prompt increased the accuracy scores of Anthropic chatbots to similar levels of other chatbots (Fig 1). ChatGPT4o had more complete answers than other chatbots (Fig 6), the quality

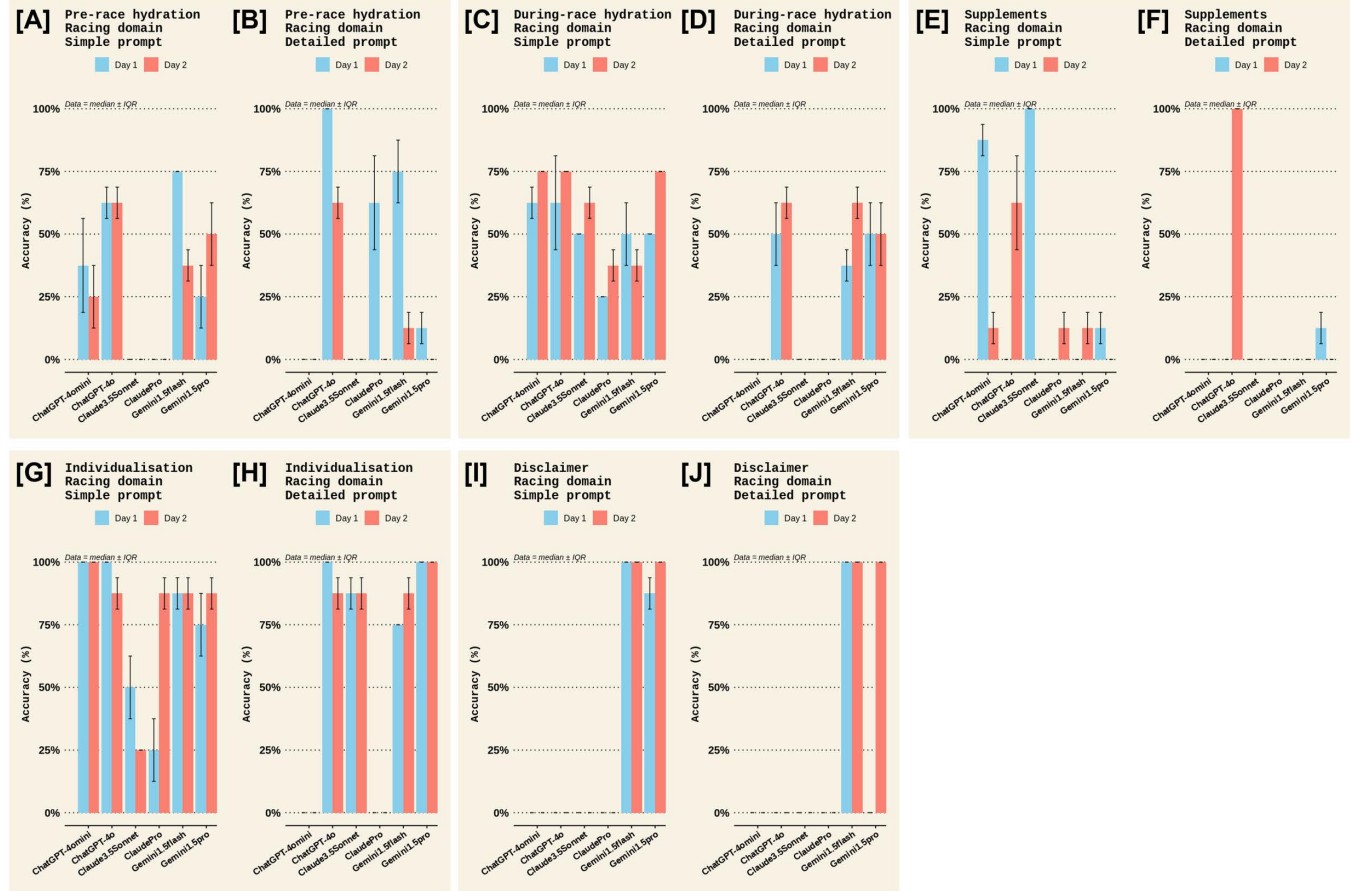

**Fig 5. Accuracy criteria in the *Racing* domain among different chatbots on the two test days in Experiment 1 (continued).** The accuracy scores for all criteria measured in the *Sports Nutrition for Racing* domain. Panels [A] and [B] show accuracy for the *Pre-race hydration* criterion for the *Simple* and *Detailed* prompts, respectively. Panels [C] and [D] show accuracy for the *During-race hydration* criterion for the *Simple* and *Detailed* prompts, respectively. Panels [E] and [F] show accuracy for the *Supplements* criterion for the *Simple* and *Detailed* prompts, respectively. Panels [G] and [H] show accuracy for the *Individualisation* criterion for the *Simple* and *Detailed* prompts, respectively. Panels [I] and [J] show accuracy for the *Disclaimer* criterion for the *Simple* and *Detailed* prompts, respectively. Each criteria score is the median ± interquartile range (IQR) of the two raters' scores; therefore, no statistical analyses could be made.

of additional information was generally moderate for all chatbots (i.e., the chatbots generally did not fabricate information; Fig 9), and all chatbots' answers had high clarity (Fig 7). Completeness, clarity, and additional info quality were unaffected by the type of prompt (*simple* vs *detailed*). However, the quality of cited evidence was rated as low for all chatbots when a simple prompt was posed but improved to moderate to high when a detailed prompt was used (Fig 8). *Experiment 2* found that Gemini1.5pro and Claude3.5Sonnet had superior accuracy in answering exam questions compared to most of the other chatbots (Fig 10). In both experiments, accuracy scores did not differ between test days, demonstrating that test-retest reliability was acceptable within a 1-week timeframe.

When examining the accuracy scores in Experiment 1 more closely (Figs 2 and 3), in the *Sports nutrition for Training* domain, most chatbots scored highly for daily carbohydrate intake, daily protein intake, and individualisation; moderately for post-session carb intake, post-session protein intake, and hydration; and, poorly for information on daily energy availability and supplements. In the *Sports nutrition for Racing* domain (Figs 4 and 5), most chatbots scored highly for information on carbohydrate intake but poorly to moderately for information on hydration and supplements. In general, chatbots

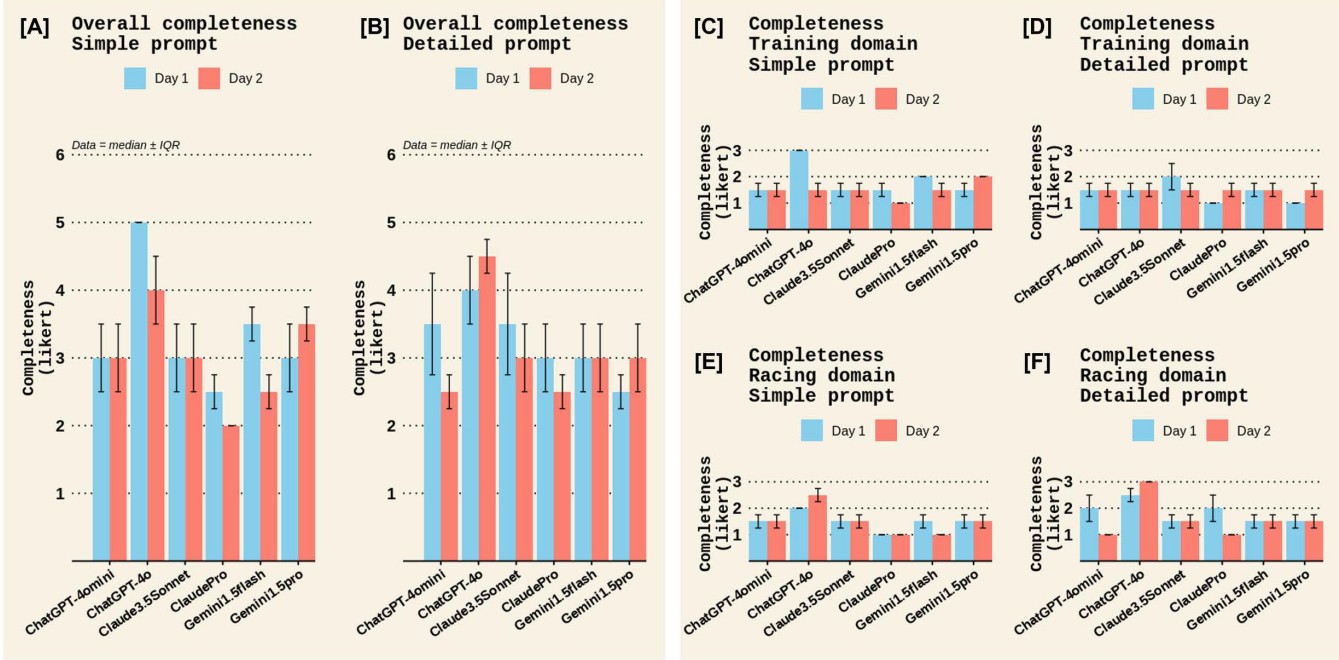

**Fig 6. Completeness among different chatbots on the two test days in Experiment 1.** Panels [A] and [B] display the overall completeness scores across both domains (*Sports Nutrition for Training* and *Sports Nutrition for Racing*) for the *Simple* and *Detailed* prompts, respectively. Panels [C] and [D] display the completeness scores in the *Training* domain for the *Simple* and *Detailed* prompts, respectively, while panels [E] and [F] show the completeness scores in the *Racing* domain. Completeness in each domain was rated on a Likert scale of 1–3; therefore, overall completeness had a maximum Likert score of 6. Data represent the median ± interquartile range (IQR) of the two raters' scores.

provided statements about using an individualised approach to nutrition but several chatbots failed to recommend seeking advice from a professional, such as a registered nutritionist/dietician. That said, ChatGPT, Claude, and Gemini include generic disclaimer statements on every output (e.g., "*ChatGPT can make mistakes. Check important info.*", "*Claude can make mistakes. Please double-check responses.*", and "*Gemini can make mistakes, including about people, so double-check it.*"). Plus, during sign-up, Claude instructs users that "*Claude is not intended to give advice, including legal, financial, & medical advice. Don't rely on our conversation alone without doing your own independent research.*".

It was hypothesised that AI chatbots could provide high-quality (accurate, complete, clear/coherent, and reliable) sports nutrition information supported by high-quality evidence. Although the findings show acceptable reliability, the accuracy and completeness of the chatbots' outputs were not exceptional. The accuracy scores ranged from 74% (Gemini1.5pro) to 31% (ClaudePro) and 89% (Claude3.5Sonnet) to 61% (ClaudePro) in Experiments 1 and 2, respectively. In general, it is surprising that chatbots did not perform better, particularly when previous studies have shown that chatbots like ChatGPT do exceptionally well on medical school exams [75–78]. However, recent evidence shows that LLMs struggle to provide accurate answers to consumer health questions [79]. Perhaps, sports nutrition is too specialised, too nuanced, or too confounded with contradictory information on the internet for success, and, therefore, LLMs need to be fine-tuned on datasets specifically curated for athletes' sports nutrition questions. One approach to improve chatbot performance could be to train LLMs on evidence-based sports nutrition guidelines combined with retrieval-augmented generation (RAG) fine-tuning by domain expert dieticians and sports nutritionists [80].

Interestingly, there was also an inconsistency in chatbot performance between the experiments. For example, why did Claude3.5Sonnet do so well on the MCQ exam in experiment 2 but so poorly when having to "think" and provide dietary

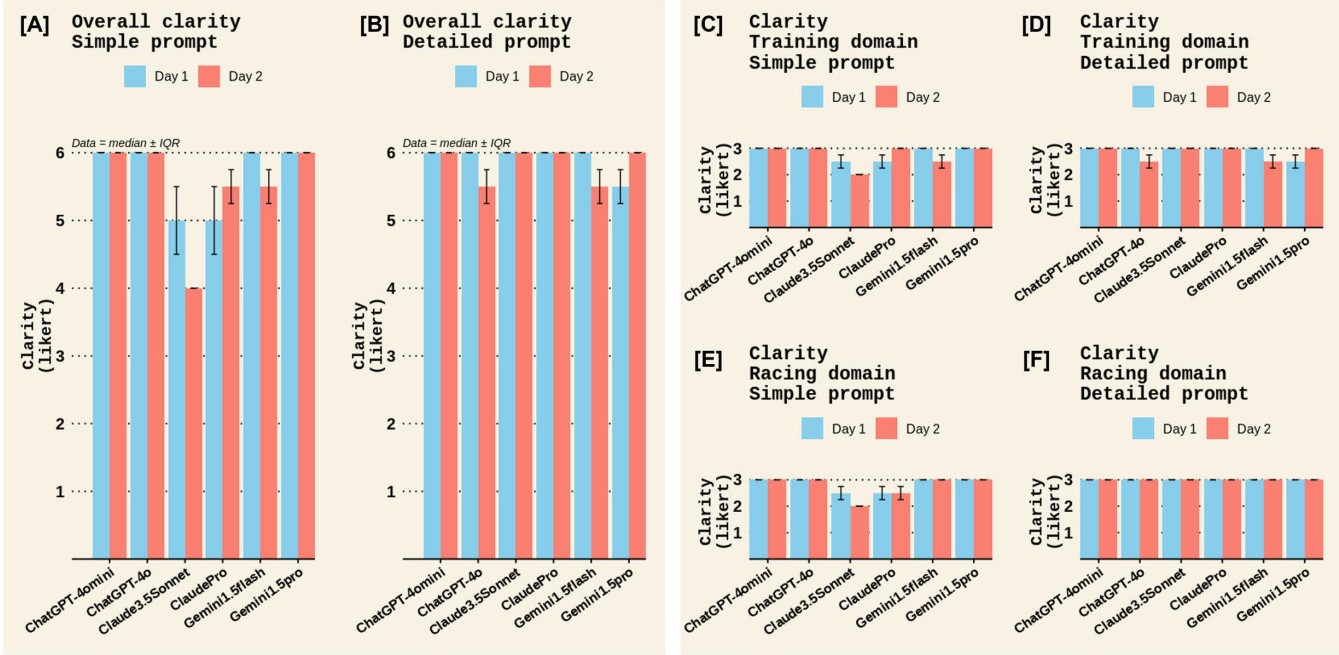

**Fig 7. Clarity among different chatbots on the two test days in Experiment 1.** Panels [A] and [B] display the overall clarity scores across both domains (*Sports Nutrition for Training* and *Sports Nutrition for Racing*) for the *Simple* and *Detailed* prompts, respectively. Panels [C] and [D] display the clarity scores in the *Training* domain for the *Simple* and *Detailed* prompts, respectively, while panels [E] and [F] show the clarity scores in the *Racing* domain. Clarity in each domain was rated on a Likert scale of 1–3 therefore, overall clarity had a maximum Likert score of 6. Data represent the median ± interquartile range (IQR) of the two raters' scores.

advice in experiment 1? And, why did Claude3.5Sonnet dramatically outperform ClaudePro in Experiment 2? Without knowing exactly how the models were trained, it is impossible to answer such questions but, in general, the findings show that the current versions of AI chatbots do not provide high-quality sports nutrition information when challenged in the zero-shot prompt manner used in this study. An obvious follow-up is to use a multi-shot "conversational" prompt approach to determine whether the chatbots' accuracy can be improved, but that is beyond the scope of this study.

One reason for the between-chatbot divergence in accuracy is how chatbots' LLMs have been trained. For example, when the prompts in this study were entered, ChatGPT and Claude did not use web scraping (they did not have access to the internet) and provided answers derived solely from the data on which their LLMs had been trained. On the other hand, Gemini provided answers derived from a combination of the data its LLM had been trained on and the internet (through web-scraping via Google Search). That said, the precise information on which chatbots' LLMs have been trained is unknown. To explain the between-chatbot divergence in Experiment 2, the best-performing chatbots may have seen a greater proportion of the exam questions during their LLM training phases. However, in experiment 1, the Claude chatbots only scored more poorly when a simple prompt was entered; a detailed prompt improved accuracy for Claude but not for ChatGPT or Gemini. This could be explained by the fact that some chatbots, including ChatGPT, can create a multi-shot chain-of-thought prompt from a zero-shot single-sentence prompt by expanding the initial prompt into a sequence of logically connected intermediate reasoning steps [28–35]. Certainly, in its earlier versions, ChatGPT did not benefit as consistently from prompt complexity due to a residual bias toward text completion and limitations in processing nested or multi-part instructions [81]. By contrast, Claude was trained using a constitutional AI framework

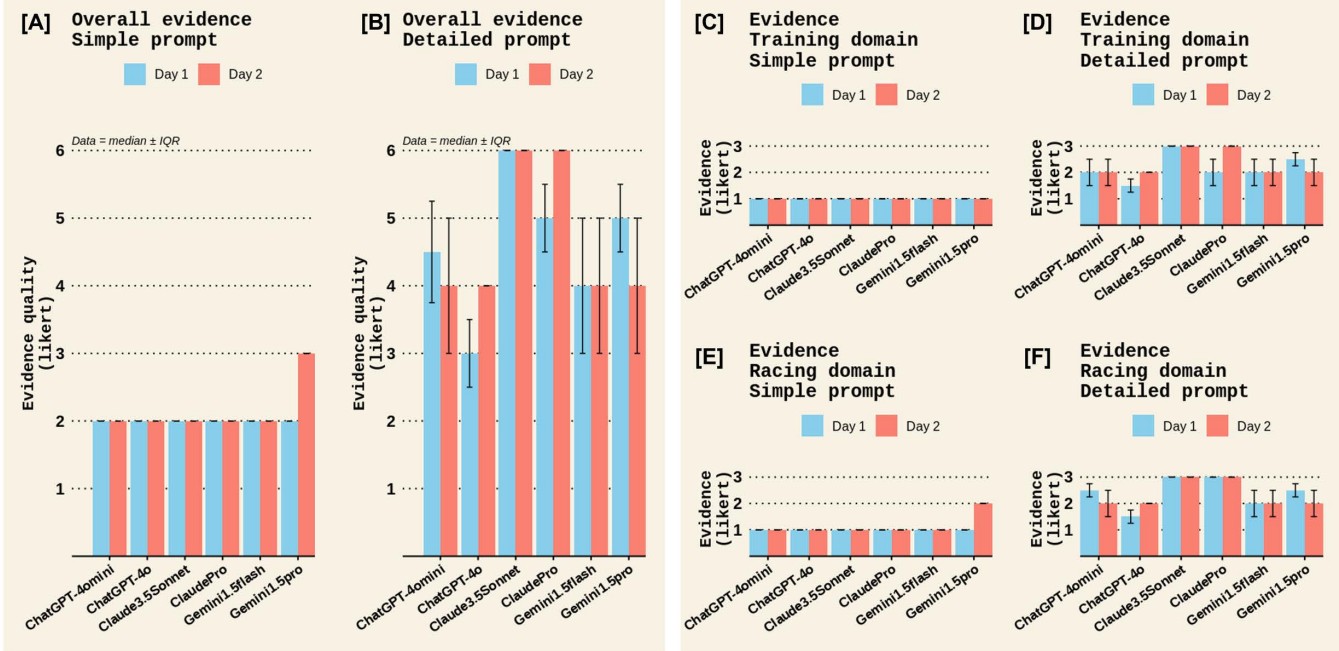

**Fig 8. The quality of cited evidence among different chatbots on the two test days in Experiment 1.** Panels [A] and [B] display the overall quality of cited evidence scores across both domains (*Sports Nutrition for Training* and *Sports Nutrition for Racing*) for the *Simple* and *Detailed* prompts, respectively. Panels [C] and [D] display the quality of cited evidence scores in the *Training* domain for the *Simple* and *Detailed* prompts, respectively, while panels [E] and [F] show the quality of cited evidence scores in the *Racing* domain. The quality of cited evidence in each domain was rated on a Likert scale of 1–3 therefore, overall evidence quality had a maximum Likert score of 6. Data represent the median ± interquartile range (IQR) of the two raters' scores.

that emphasises ethical reasoning, multi-step task parsing, and contextual sensitivity — features that are better leveraged when prompts are more specific and structured [82]. This may explain why a detailed prompt improved the accuracy of Claude compared to a simple prompt.

Furthermore, in experiment 2, all chatbots scored a higher proportion of correct answers in the *Exercise and performance nutrition* and *Nutrition operation and management* domains compared to the *Clinical sports nutrition* domain. Therefore, it is possible that the LLMs were less well trained on clinical topics like energy balance and availability, weight management, special populations, and disordered eating compared to topics found in the other two domains (e.g., energy metabolism, fuelling for training and competition, sports foods, food and beverage management, and nutrition administration). These types of issues are examples of AI training data bias and recency bias [36].

Another important point is that the chatbots provided diverse information about hydration in Experiment 1. This is unsurprising because hydration is a complicated topic with many opinions: *drink to thirst* vs. *drink to a schedule* vs. *drink to replace sweat losses* [83–86]. Older guidelines provided specific doses of fluid to consume; however, because hydration status is so greatly influenced by training intensity, environmental conditions, body mass, baseline hydration status, etc, most guidelines now suggest that a well-practised hydration approach should either include a drink-to-thirst approach or a schedule that limits sweat losses but never exceeds them [14–16]. Consequently, chatbots have probably collected their knowledge of hydration from an array of sources, creating a range of diverse answers. On the contrary, the high accuracy for carbohydrate intake was less surprising because guidelines on this topic are very similar between sources — in the range of 30–90 g per hour during exercise and 3–12 g/kg/day for daily intake [14–16].

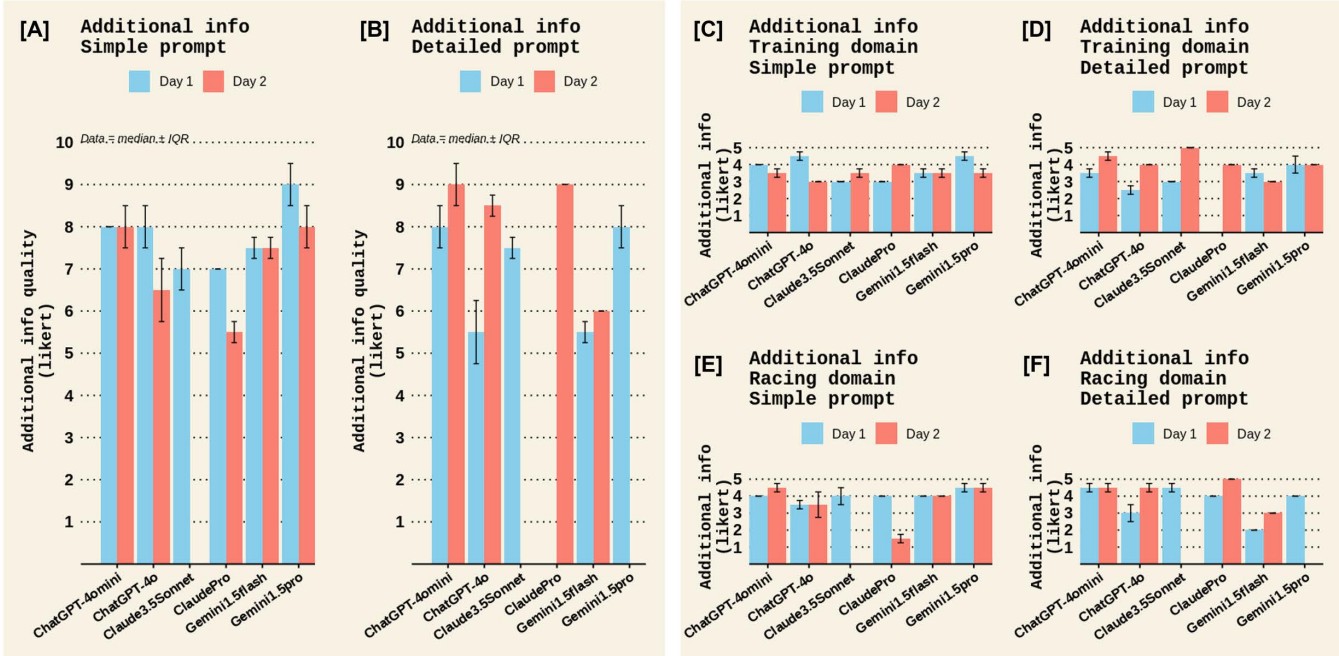

**Fig 9. The quality of additional information among different chatbots on the two test days in Experiment 1.** Panels [A] and [B] display the overall quality of additional information scores across both domains (*Sports Nutrition for Training* and *Sports Nutrition for Racing*) for the *Simple* and *Detailed* prompts, respectively. Panels [C] and [D] display the quality of additional information scores in the *Training* domain for the *Simple* and *Detailed* prompts, respectively, while panels [E] and [F] show the quality of additional information scores in the *Racing* domain. The quality of additional information in each domain was rated on a Likert scale of 1–5 therefore, overall additional information quality had a maximum Likert score of 10. Data represent the median ± interquartile range (IQR) of the two raters' scores.

**Table 6. Chatbot accuracy scores in Experiment 2.**

| Chatbot | Test day | Accuracy scores | | Contrasts |
|---|---|---|---|---|
| | | mean | SE | |
| ChatGPT-4omini | Day1 | 73% | 4% | ae |
| ChatGPT-4omini | Day2 | 70% | 4% | |
| ChatGPT-4o | Day1 | 77% | 4% | b |
| ChatGPT-4o | Day2 | 80% | 4% | |
| Claude3.5Sonnet | Day1 | 89% | 3% | abcd |
| Claude3.5Sonnet | Day2 | 88% | 3% | |
| ClaudePro | Day1 | 74% | 4% | cf |
| ClaudePro | Day2 | 61% | 5% | |
| Gemini1.5flash | Day1 | 69% | 4% | dg |
| Gemini1.5flash | Day2 | 72% | 4% | |
| Gemini1.5pro | Day1 | 85% | 3% | efg |
| Gemini1.5pro | Day2 | 86% | 3% | |

SE = standard error

Claude3.5Sonnet scored a higher proportion of correct answers to the exam than ChatGPT-4omini (comparison "a": p = 0.0001; r = −0.976, 95%CI −0.997 to −0.792), ChatGPT-4o ("b": p = 0.04; r = −0.948, 95%CI −0.994 to −0.589), ClaudePro ("c": p < .0001; r = −0.983, 95%CI −0.998 to −0.846), and Gemini1.5flash "d": (p < .0001; r = −0.978, 95%CI −0.998 to −0.807). Gemini1.5pro also scored a higher proportion of correct answers than ChatGPT-4omini ("e": p = 0.004; r = −0.964, 95%CI −0.700 to 0.952), ClaudePro ("f": p = 0.0001; r = −0.976, 95%CI −0.998 to −0.793), and Gemini1.5flash ("g": p = 0.002; r = −0.967, 95%CI −0.997 to −0.726).

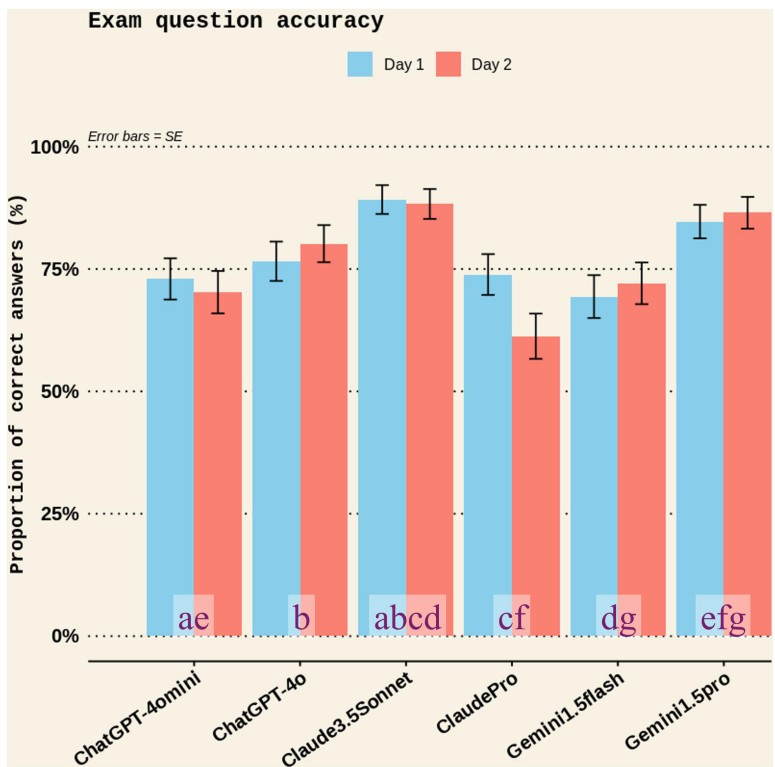

**Fig 10. The proportion of correct answers to exam questions among different chatbots on the two test days in Experiment 2.** Bars represent the proportion of correct answers and error bars represent the standard error (SE). Claude3.5Sonnet scored a higher proportion of correct answers to the exam than ChatGPT-4omini (comparison "a": p = 0.0001; r = −0.976, 95%CI −0.997 to −0.792), ChatGPT-4o ("b": p = 0.04; r = −0.948, 95%CI −0.994 to −0.589), ClaudePro ("c": p < .0001; r = −0.983, 95%CI −0.998 to −0.846), and Gemini1.5flash "d": (p < .0001; r = −0.978, 95%CI −0.998 to −0.807). Gemini1.5pro also scored a higher proportion of correct answers than ChatGPT-4omini ("e": p = 0.004; r = −0.964, 95%CI −0.700 to 0.952), ClaudePro ("f": p = 0.0001; r = −0.976, 95%CI −0.998 to −0.793), and Gemini1.5flash ("g": p = 0.002; r = −0.967, 95%CI −0.997 to −0.726). The proportion of correct answers was not different between test days.

## Limitations

This study has several limitations that should be considered when interpreting the findings.

*Selection bias:* The selection criteria for evaluating chatbot responses in Experiment 1 were based on our professional experience in sports nutrition as running coaches, along with our *a priori* knowledge of established sports nutrition guidelines, consensus statements, and position stands. Other researchers with different backgrounds might establish alternative criteria. To mitigate this limitation, in Experiment 2, the chatbots' ability to answer sports nutrition exam questions was also tested. Furthermore, we selected a limited number of chatbots; to enhance knowledge in this field, additional chatbots should be studied.

*AI bias*: Firstly, the subscription-based nature of advanced chatbot models limits accessibility to low-income populations. Secondly, the outputs in this study may have been influenced by training data bias and recency bias [36], meaning that the chatbot's LLMs may have been trained on older datasets that are missing contemporary knowledge. Such biases are amplified by the lack of transparency of the training models used by various chatbot manufacturers.

*Prompt entry problems:* During Experiment 2, the process of uploading exam questions varied across platforms. While ChatGPT allowed direct PDF uploads, neither Gemini nor Claude supported this feature. However, for Claude, the prompt character limit was sufficient for LT to paste all questions directly into its prompt box. In contrast, Gemini required splitting

**Table 7. Contrasts for ChatbotID in Experiment 2.**

| Contrast | Z-ratio | P-value | | r | 95%CI | | Power |
|---|---|---|---|---|---|---|---|
| ClaudePro – Claude3.5Sonnet | −5.330 | <.0001 | *** | −0.983 | −0.998 | −0.846 | 1.000 |
| Gemini1.5flash – Claude3.5Sonnet | −4.690 | <.0001 | *** | −0.978 | −0.998 | −0.807 | 0.997 |
| ClaudePro – Gemini1.5pro | −4.504 | 0.0001 | *** | −0.976 | −0.998 | −0.793 | 0.995 |
| ChatGPT-4omini – Claude3.5Sonnet | −4.501 | 0.0001 | *** | −0.976 | −0.997 | −0.792 | 0.994 |
| Gemini1.5flash – Gemini1.5pro | −3.825 | 0.002 | ** | −0.967 | −0.997 | −0.726 | 0.969 |
| ChatGPT-4omini – Gemini1.5pro | −3.624 | 0.004 | ** | −0.964 | −0.996 | −0.700 | 0.952 |
| ChatGPT-4o – Claude3.5Sonnet | −2.965 | 0.036 | * | −0.948 | −0.994 | −0.589 | 0.842 |
| ChatGPT-4o – ClaudePro | 2.628 | 0.091 | | 0.935 | 0.510 | 0.993 | 0.748 |
| ChatGPT-4o – Gemini1.5pro | −2.011 | 0.336 | | −0.895 | −0.989 | −0.307 | 0.520 |
| Gemini1.5flash – ChatGPT-4o | −1.901 | 0.401 | | −0.885 | −0.987 | −0.261 | 0.477 |
| ChatGPT-4o – ChatGPT-4omini | 1.688 | 0.540 | | 0.860 | 0.162 | 0.985 | 0.393 |
| Gemini1.5pro – Claude3.5Sonnet | −1.009 | 0.915 | | −0.710 | −0.965 | 0.239 | 0.172 |
| ClaudePro – ChatGPT-4omini | −0.958 | 0.931 | | −0.692 | −0.963 | 0.273 | 0.160 |
| Gemini1.5flash – ClaudePro | 0.742 | 0.977 | | 0.596 | −0.417 | 0.949 | 0.115 |
| Gemini1.5flash – ChatGPT-4omini | −0.216 | 1.000 | | −0.211 | −0.873 | 0.725 | 0.055 |

CI = confidence interval of r; *, **, and *** indicate significance at 5%, 1%, and 0.1% levels.

**Table 8. Contrasts for ExamDomain in Experiment 2.**

| Contrast | Z-ratio | P-value | | r | 95%CI | | Power |
|---|---|---|---|---|---|---|---|
| Exercise and performance nutrition – Clinical sports nutrition | 7.736 | <.0001 | *** | 0.992 | 0.977 | 0.997 | 1.000 |
| Clinical sports nutrition – Nutrition operation and management | −3.783 | 0.0005 | *** | −0.967 | −0.988 | −0.911 | 0.966 |
| Exercise and performance nutrition – Nutrition operation and management | −0.510 | 0.866 | | −0.454 | −0.760 | 0.016 | 0.080 |

CI = confidence interval of r; *** indicates significance at 0.1% level.

**Table 9. Contrasts for TestDay in Experiment 2.**

| Contrast | Z-ratio | P-value | r | 95%CI | | Power |
|---|---|---|---|---|---|---|
| TestDay1 – TestDay2 | 0.614 | 0.539 | 0.523 | −0.072 | 0.844 | 0.094 |

CI = confidence interval of r

the exam into two separate prompts due to its character limitations. These differences in prompt management could have influenced chatbot performance.

*Lack of conversation:* Experiment 1 evaluated each chatbot's response based on a single prompt without follow-up interactions. In practical use, users typically engage in multi-turn conversations, refining answers through chain-of-thought prompting. Therefore, our findings might underestimate the chatbots' potential accuracy under interactive conditions. That said, a user-directed conversation could "influence" or "force" specific answers from a chatbot that align with a user's pre-conceived biases on a topic.

*Chatbot memory management:* At the time of data collection, ChatGPT allowed users to delete conversation histories and disable settings that improved the model with user input. Claude operated without retaining personal conversation histories. Conversely, Gemini stored conversation histories for up to 72 hours, even when its activity tracking was disabled (when Gemini Apps Activity was turned off). Although manual deletion was possible through Google's settings (at myactivity.google.com/product/gemini), it was unclear whether this fully prevented Gemini's LLM learning from past prompts [87]. Consequently, memory-related learning effects could have influenced test-retest reliability in Gemini's performance.

*Transparency:* The transparency of chatbot development and updates varies among providers. While OpenAI and Anthropic maintain relatively clear update protocols, Google's Gemini offer less transparency regarding their training processes and update cycles. This limited visibility into LLM development complicates the assessment of their evolving capabilities.

*Statistical power:* Between-chatbot comparisons in Experiment 1 revealed moderate effect sizes but low to moderate statistical power. This limited power heightened the risk of Type II errors, meaning that potential differences might have gone undetected. Therefore, Experiment 1 was underpowered to conclusively detect performance differences between chatbots and would benefit from a larger number of criteria. Using the effect sizes in Table 5, we estimate that using 35 criteria, rather than the original 20, would achieve at least 80% power to detect differences between 6 chatbots at a 5% level of statistical significance.

*Rapid chatbot development*: Given the rapid pace of LLM advancement, chatbots' capabilities are expected to improve significantly beyond the performance levels observed in this study. As such, our findings may become outdated quickly and future re-tests may yield different results. Nonetheless, our study design and results provide a useful framework for future research assessing the accuracy and test-retest reliability of AI chatbots in sports nutrition and related fields.

## Strengths and future directions

Although some studies have investigated chatbots' ability to prescribe training [17–19] and diet plans [20–24], many studies have only used qualitative approaches to assess accuracy [18–21] and few studies have compared the accuracy of different chatbots [17,18,23]. A major strength of this study is minimizing bias through several approaches: using an independent prompt inputter (LT), the blinding of investigators to the chatbot ID throughout the chatbot rating and data analysis steps, the independent ratings by the two investigators (Experiment 1), using an automated script to rate the exam questions (Experiment 2), deleting chatbot history, deactivating chatbot memory, and creating a new chat for every new prompt entry. Another major strength of this study is the quantitative methods used to compare the accuracy and test-retest reliability of basic and advanced chatbot models from OpenAI, Google, and Anthropic, who have independently developed distinct LLMs. In doing so, the most appropriate statistical model was chosen based on chi-squared *goodness of fit* testing and AIC/BIC prediction error assessments. Furthermore, completeness, clarity, and evidence quality were also assessed to provide a more holistic view of chatbot performance. This study also used best practice prompting techniques to develop detailed prompts based on the prompt engineering principles used to train LLMs [28–35]. Doing so helped improve the accuracy of some chatbots (Claude3.5Sonnet and Claude Pro). Additionally, although there was a high risk of a Type II error in Experiment 1, between-chatbot comparisons in Experiment 2 had large effect sizes and high statistical power, suggesting a low risk of false negative findings. Consequently, the effect sizes generated by this study can inform sample sizes in future studies to ensure adequate power. Finally, the methodology used in this study will also help future studies use quantified approaches to assess the accuracy and test-retest reliability of AI chatbots. Standardised guidelines for using generative AI chatbots are urgently needed if chatbots are to be trusted to advance science and clinical/coaching practice [88,89]. The information produced by this study will help contribute towards creating such guidelines.

## Conclusion

This study demonstrates that while the test-retest reliability of ChatGPT, Claude, and Gemini is acceptable, their performance in providing high-quality sports nutrition information when given a zero-shot prompt is questionable. In general, all the chatbots tested had a good understanding of the basic concepts of sports nutrition but lacked knowledge of more nuanced areas like energy availability, hydration, and supplements. So, although chatbots have the potential to act as a "copilot" for advancing an athlete's/coach's knowledge, this potential will only be fulfilled if said athlete/coach can trust that the chatbot has domain expertise and/or if an athlete's/coach's domain expertise is sufficient to identify incorrect or incomplete information. Therefore, until the LLMs underpinning these generative AI chatbots are improved, an athlete or coach seeking tailored sports nutrition advice should seek professional input from a registered nutritionist/dietician.

## Supporting information

**S1 File.** Supplemental Tables for Statistical Analysis. **S1 Table.** The ANOVA summary table for Experiment 1. **S2 Table.** The partial eta-squared values for the ANOVA model for Experiment 1. **S3 Table.** Pairwise comparisons (inc. Cohen's d effect sizes) for ChatbotID in the ANOVA model for Experiment 1. **S4 Table.** Pairwise comparisons (inc. Cohen's d effect sizes) for TestDay in the ANOVA model for Experiment 1. **S5 Table.** Pairwise comparisons (inc. Cohen's d effect sizes) for Domain in the ANOVA model for Experiment 1. **S6 Table.** Pairwise comparisons (inc. Cohen's d effect sizes) for PromptType in the ANOVA model for Experiment 1. **S7 Table.** Pairwise comparisons (inc. Cohen's d effect sizes) for the ChatbotID * Prompt_type interaction in the ANOVA model for Experiment 1. **S8 Table.** The LMM model summary table for Experiment 2. **S9 Table.** Pairwise comparisons for ChatbotID in the LMM model for Experiment 2. **S10 Table.** Pairwise comparisons for ExamDomain in the LMM model for Experiment 2. **S11 Table.** Pairwise comparisons for TestDay in the LMM model for Experiment 2. **S12 Table.** Effect sizes (r) and power for ChatbotID in the LMM model for Experiment 2. **S13 Table.** Effect sizes (r) and power for ExamDomain in the LMM model for Experiment 2. **S14 Table.** Effect sizes (r) and power for TestDay in the LMM model for Experiment 2.
(DOCX)

**S2 File.  METRICS Checklist.**
(DOCX)

**S3 File.  GRRAS Checklist.**
(DOCX)

## Acknowledgments

The authors thank Ronnie Harper at My Sports Dietitian for providing the exam questions used to prepare candidates for the Certified Specialist in Sports Dietetics (CSSD) board exam (https://mysportsd.com/cssd-study-guide). The authors also thank Lisa Tindle for overseeing the blinding process, chatbot prompt entry, and chatbot output retrieval.

## Author contributions

**Conceptualization:** Matthew J. Laye, Thomas P.J. Solomon.

**Data curation:** Thomas P.J. Solomon.

**Formal analysis:** Thomas P.J. Solomon.

**Methodology:** Matthew J. Laye, Thomas P.J. Solomon.

**Writing – original draft:** Matthew J. Laye, Thomas P.J. Solomon.

**Writing – review & editing:** Matthew J. Laye, Thomas P.J. Solomon.

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
