## [Decision Letter · Decision Letter 0]

4 Apr 2025

Dear Dr. Laye,

Thank you for submitting your manuscript to PLOS ONE. After careful consideration, we feel that it has merit but does not fully meet PLOS ONE’s publication criteria as it currently stands. Therefore, we invite you to submit a revised version of the manuscript that addresses the points raised during the review process.

We look forward to receiving your revised manuscript.

Kind regards,

Sirwan Khalid Ahmed

Academic Editor

PLOS ONE

Journal Requirements:

2. Thank you for stating the following in the Competing Interests section: [TS has given invited talks at societal conferences and university/pharmaceutical symposia for which the organisers paid for travel and accommodation; he has also received research money from publicly funded national research councils and medical charities, and private companies, including Novo Nordisk Foundation, AstraZeneca, Amylin, AP Møller Foundation, and Augustinus Foundation; and, he has consulted for Boost Treadmills, GU Energy, and Examine.com, and owns a consulting business, Blazon Scientific, and an endurance athlete education business, Veohtu. These companies have had no control over the research design, data analysis, or publication outcomes of this work.  ML has given invited talks at societal conferences and university symposia and meetings for which the organisers paid for travel and accommodation; he has received research money from Augustinus Foundation, American College of Sports Medicine, and national research institutions; and, he has consulted for Zepp Health, Levels Health, GU Energy, and EAB labs, and has coached for Sharman Ultra Coaching. These companies have had no control over the research design, data analysis, or publication outcomes of this work.  My Sports Dietitian provided a set of multiple-choice questions designed to resemble the Certified Specialist in Sports Dietetics (CSSD) board exam. Neither TPJS nor MJL have any financial relationships with My Sports Dietitian.].

We note that you received funding from a commercial source: [Novo Nordisk Foundation, AstraZeneca, Amylin, AP Møller Foundation, and Augustinus Foundation].

3. Thank you for stating the following financial disclosure: [The work was funded by internal institutional funds from the Idaho College of Osteopathic Medicine. No external funds were received for this work.]. 

Reviewers' comments:

Reviewer's Responses to Questions

**Comments to the Author**

1. Is the manuscript technically sound, and do the data support the conclusions?

Reviewer #1: Partly

Reviewer #2: Partly

Reviewer #3: Yes

2. Has the statistical analysis been performed appropriately and rigorously?

Reviewer #1: Yes

Reviewer #2: I Don't Know

Reviewer #3: Yes

3. Have the authors made all data underlying the findings in their manuscript fully available?

Reviewer #1: Yes

Reviewer #2: Yes

Reviewer #3: Yes

4. Is the manuscript presented in an intelligible fashion and written in standard English?

Reviewer #1: Yes

Reviewer #2: No

Reviewer #3: Yes

Reviewer #1: 1) The “basic” and “advanced” versions of AI chatbots are clearly categorized. However, additional information should be provided about the version dates of the LLM models and the datasets they were trained on.

2) The simple and detailed prompts entered into the chatbots are clearly listed. However, is there a control mechanism on the update status of the chatbots or the impact of users’ past chat history? Please provide some information about this.

3) More information can be added about the data collection process and bias control.

4) The time between two tests is very short (4 days: October 7, 2024 and October 11, 2024). Since AI models can be updated frequently, a longer time interval may be required for test-retest.

5) The study stated that the referees who evaluated the chatbot responses evaluated the outputs blindly. However: The evaluation criteria used by the referees were not clearly given. The criteria used by the referees for scoring should be explained in more detail and compared with other referees.

6) It should be detailed which types of nutrition questions each model is more successful in. For example, if Gemini uses more scientific references and Claude explains in more general terms, this should be emphasized.

Reviewer #2: The study was well-designed and the statistics are good. However, some points should be addressed.

the first page of introduction may be deleted. Kindly focus on three elements of introduction.

a. What is known about the topic? (Background)

b. What is not known? (The research problem)

c. Why the study was done? (Justification)

methods

Model

what are the exact settings for each tool?

Evaluation

What is the exact approach used to evaluate the content generated by the generative AI-based model and is it an objective or subjective evaluation?

Explain in details the Randomisation

Was the process of selecting the topics to be tested on the generative AI-based model randomized?

Specificity is not clear ,please clarify the following points

How specific are the exact prompts used?

Were those exact prompts provided fully?

Did the authors consider the feedback and learning loops?

How specific are the language and cultural issues considered in the generative AI model?

Discussion:

1. The discussion section needs to be described scientifically. Kindly frame it along the following lines:

i. Main findings of the present study

ii. Comparison with other studies

iii. Implication and explanation of findings

iv. Strengths and limitations

v. Conclusion, recommendation

Reviewer #3: ADDITIONAL COMMENTS FOR THE AUTHOR

1. Title, Abstract and Introduction

• Lines 8 – 14 lacks specific details about the statistical methods used to assess chatbot performance. Also, statistical methods for assessing interrater reliability and chatbot accuracy should be mentioned.

• Lines 24 – 25 could be extended to include suggestions or recommendations for improving chatbot reliability.

• In lines 50 – 58, chatbot variability was mentioned but it does not explicitly discuss potential biases in AI-generated responses. It is important to address that.

• Details on how prompts were standardized across chatbots (lines 72-74) would improve replicability.

• Potential biases in chatbot outputs (lines 75-76) should be explicitly acknowledged.

• Recommendations for improving chatbot-generated sports nutrition advice (lines 108 - 110) should be included.

2. Study Design (Lines 80 – 83)

• It would be beneficial to explicitly mention any potential biases introduced by modifications to the original protocol. Clearly state what amendments were made post-registration and their potential impact on findings.

3. Table 2 - Chatbots Used (Line 129)

• The subscription-based nature of advanced models could influence accessibility and should be discussed as a potential limitation.

4. Assumptions for Likert Scale Data (Lines 264 – 273)

• The justification for treating Likert scores as continuous when aggregated is common but should be explicitly supported by references or sensitivity analyses.

• The log transformation of the Likert scores is reasonable, but additional justification is recommended to address concerns about its interpretability.

5. Interpretation of Accuracy Categories (Lines 305 – 327)

• The categorical thresholds for percentage accuracy scores are well defined but it is unclear if these thresholds were pre-specified or data-driven, which could introduce bias.

6. Clarity and Readability (lines 529 – 539)

• Some statistical results are presented in a dense manner that may be difficult for a general audience to interpret.

• Consider summarizing key findings in a more reader-friendly format (e.g., adding bullet points).

7. Use of Figures and Tables

• Table 5 includes multiple p-values; consider highlighting significant results for readability (Line 396).

8. Experiment 2 Model Justification

• The logistic mixed model was chosen based on AIC/BIC (Lines 506 – 509), but additional justification on model assumptions (e.g., normality of residuals) would be beneficial.

• The random intercept for ChatbotID was reported as zero (Line 509 - 510), but more details on its implications should be provided.

9. Grammar and Style Adjustments

• Consider rewording the quoted sentence for clarity: “The comparisons had moderate effect sizes but only low to moderate statistical power" (Line 370 – 371).

• Consider changing "are" to "is" in the quoted excerpt: "Full post hoc data for the ChatbotID × PromptType interaction are available..." (Lines 380 - 381).

10. Clarity and Justification of Key Findings

• The distinction between "accuracy," "completeness," and "clarity" should be defined earlier to prevent reader confusion.

• Some comparative claims about chatbot performance lack direct references to quantitative results (e.g., Lines 552 – 553: “ChatGPT4o had more complete answers”).

• The explanation for why chatbots struggled with certain topics (e.g., hydration, energy availability) should be expanded with references to chatbot training limitations.

11. Statistical Power and Methodological Transparency

• Experiment 1 is noted as underpowered (Lines 661 – 662). The authors should provide details on the required sample size to detect meaningful differences.

12. Implications for AI Development

• While the manuscript highlights the need for chatbot fine-tuning, it does not offer specific recommendations for AI developers. Adding this would enhance the practical impact of the study.

**Do you want your identity to be public for this peer review?** For information about this choice, including consent withdrawal, please see our Privacy Policy

Reviewer #1: **Yes: ** Esedullah AKARAS

Reviewer #2: No

Reviewer #3: **Yes: ** Ojore Godday Aghedo

---

## [Author Response · Author response to Decision Letter 1]

24 Apr 2025

We have attached a document outlining our responses to the reviewers. We have repeated that material here.

Response to Reviewers

Editor’s comments:

Thank you for your comments. We have addressed them point-by-point in bold blue font below.

Our revised manuscript is formatted according to style requirements outlined in the weblinks.

2. Thank you for stating the following in the Competing Interests section: [TS has given invited talks at societal conferences and university/pharmaceutical symposia for which the organisers paid for travel and accommodation; he has also received research money from publicly funded national research councils and medical charities, and private companies, including Novo Nordisk Foundation, AstraZeneca, Amylin, AP Møller Foundation, and Augustinus Foundation; and, he has consulted for Boost Treadmills, GU Energy, and Examine.com, and owns a consulting business, Blazon Scientific, and an endurance athlete education business, Veohtu. These companies have had no control over the research design, data analysis, or publication outcomes of this work. ML has given invited talks at societal conferences and university symposia and meetings for which the organisers paid for travel and accommodation; he has received research money from Augustinus Foundation, American College of Sports Medicine, and national research institutions; and, he has consulted for Zepp Health, Levels Health, GU Energy, and EAB labs, and has coached for Sharman Ultra Coaching. These companies have had no control over the research design, data analysis, or publication outcomes of this work. My Sports Dietitian provided a set of multiple-choice questions designed to resemble the Certified Specialist in Sports Dietetics (CSSD) board exam. Neither TPJS nor MJL have any financial relationships with My Sports Dietitian.]. We note that you received funding from a commercial source: [Novo Nordisk Foundation, AstraZeneca, Amylin, AP Møller Foundation, and Augustinus Foundation]. Please provide an amended Competing Interests Statement that explicitly states this commercial funder, along with any other relevant declarations relating to employment, consultancy, patents, products in development, marketed products, etc. Within this Competing Interests Statement, please confirm that this does not alter your adherence to all PLOS ONE policies on sharing data and materials by including the following statement: "This does not alter our adherence to PLOS ONE policies on sharing data and materials.” (as detailed online in our guide for authors http://journals.plos.org/plosone/s/competing-interests). If there are restrictions on sharing of data and/or materials, please state these. Please note that we cannot proceed with consideration of your article until this information has been declared. Please include your amended Competing Interests Statement within your cover letter. We will change the online submission form on your behalf.

We have added this information to the manuscript and included the revised Competing Interests statement in our cover letter.

3. Thank you for stating the following financial disclosure: [The work was funded by internal institutional funds from the Idaho College of Osteopathic Medicine. No external funds were received for this work.]. Please state what role the funders took in the study. If the funders had no role, please state: ""The funders had no role in study design, data collection and analysis, decision to publish, or preparation of the manuscript."" If this statement is not correct you must amend it as needed. Please include this amended Role of Funder statement in your cover letter; we will change the online submission form on your behalf.

We have added this information to the manuscript and included this statement in our cover letter.

Reviewers' comments:

Reviewer 1

Thank you for your comments. We have addressed them point-by-point in bold blue font below.

1. The “basic” and “advanced” versions of AI chatbots are clearly categorized. However, additional information should be provided about the version dates of the LLM models and the datasets they were trained on.

We have added release date information to Table 2.

The model manufacturers do not provide precise details on which datasets their models are trained on, but they are all multimodal models, meaning they were trained on several datasets. Importantly, we did not fine-tune the models in any way; we added information to clarify that on lines 118 and 220.

2. The simple and detailed prompts entered into the chatbots are clearly listed. However, is there a control mechanism on the update status of the chatbots or the impact of users’ past chat history? Please provide some information about this.

Yes, there was a control mechanism for the update status of the chatbots: We input the test and retest prompts on October 7, 2024 and October 11, 2024, less than 1 week apart, to prevent model updates influencing changes in the output. No model updates were made in that timeframe; we added information to clarify that point on lines 121 and 233.

And, yes, we also considered the impact of users’ past chat history. For example, as explained in the Prompt inputting sections (lines 115 and 226), new chatbot accounts were created, the chatbots’ histories were deleted before each new prompt was entered, the chatbots’ memories were turned off, and a new chat was created each time a new prompt was entered.

3. More information can be added about the data collection process and bias control.

The original manuscript described the data collection process with a high level of detail. For example, the Prompt inputting sections (lines 125-129 and lines 244-247) describe how our independent prompter entered the prompts and collected and blinded the data. However, although the processes were described, their purpose (i.e., to minimise bias) was not explicitly stated. So, please see that we have added some text to those sections to clarify that these approaches were used to minimise bias.

Other approaches to minimise bias included: the independent ratings by the two investigators (Experiment 1: line 145-146), the use of an automated script to rate the exam questions (Experiment 2: line 251-252), the deletion of chatbot history (lines 122-124), not using chatbot customisations or plug-ins (lines 122-124), turning the chatbot’s memory off (lines 122-124), and creating a new chat each time a new prompt was entered (lines 122-124). We have added some text to lines 698-700 to clarify these points.

Furthermore, we used the intraclass correlation coefficient (ICC) to examine absolute agreement between the raters’ scores (lines 273-376), finding that the interrater reliability between the two raters was good (ICC = 0.893, 95% CI 0.869 to 0.912; F = 9.82, p < 0.001) (line 349). Additionally, we used the Kuder-Richardson 20 (KR-20) statistic to measure the internal consistency reliability of the exam (line 308-311), finding a KR-20 value of 0.895, indicating good internal reliability of the exam and suggesting that the exam questions were measuring the same underlying construct consistently and, thereby, effectively assessing the knowledge it was designed to measure (line 513-515).

4. The time between two tests is very short (4 days: October 7, 2024 and October 11, 2024). Since AI models can be updated frequently, a longer time interval may be required for test-retest.

If our research question was to examine the time course of how chatbot outputs change with time (i.e. with model development), then a longer time interval would have been prudent. However, the purpose of the test-retest approach in the context of our research question was to examine the test-retest reliability of information provided by chatbots. If a longer time interval was chosen, a chatbot model may have been updated (because AI models are updated frequently); consequently, any comparisons between dates would introduce bias and be unable to provide information regarding the reliability of a specific chatbot model.

5. The study stated that the referees who evaluated the chatbot responses evaluated the outputs blindly. However: The evaluation criteria used by the referees were not clearly given. The criteria used by the referees for scoring should be explained in more detail and compared with other referees.

The evaluation criteria are described and explained in detail on lines 133-215, 269-287, and 312-318 and complemented by the information provided in Table 3. These evaluation approaches have been used previously in similar diet/exercise investigations. For example, Haman et al, Papastratis et al, Sun et al, and Niszczota et al measured the accuracy of chatbot outputs, but they only used qualitative evaluations, not statistical analyses, to examine between-chatbot comparisons. Meanwhile, other studies have used Likert scales to compare different metrics between chatbots, including accuracy (Düking et al, Pugliese et al, Pérez-Guerrero et al, Goodman et al), completeness/comprehensiveness (Pugliese et al, Pérez-Guerrero et al, Goodman et al), clarity/coherence (Pérez-Guerrero et al, Goodman et al), evidence quality (Pérez-Guerrero et al), and test-retest reliability (Goodman et al), along with statistical tests to examine between-chatbot comparisons.

While we cited all these papers in the original manuscript, we did not explicitly indicate how these evaluation approaches had previously been used. We have added text to remedy that issue (lines 148-149).

6. It should be detailed which types of nutrition questions each model is more successful in. For example, if Gemini uses more scientific references and Claude explains in more general terms, this should be emphasized.

This information was provided on line 403 onwards, but we have added a little more detail on lines 404-406 and 412-416 to address your point and improve the clarity of the results.

Reviewer 2

The study was well-designed and the statistics are good. However, some points should be addressed.

Thank you for your comments. We have responded point-by-point in bold blue text below.

The first page of introduction may be deleted. Kindly focus on three elements of introduction: a. What is known about the topic? (Background). b. What is not known? (The research problem). c. Why the study was done? (Justification)

These are general comments on how to write an introduction section. Our original introduction already contains clear information about the background, the problem, and the justification. We will not delete the first page.

Model - What are the exact settings for each tool?

Our original methods section already contains this information. Please see Table 2 and the Prompt inputting sections on lines 115 and 226.

Evaluation - What is the exact approach used to evaluate the content generated by the generative AI-based model and is it an objective or subjective evaluation?

Our original methods section already contains this information. Please see the Rating chatbot performance sections on lines 133-211 and 249-254. We have added info on lines 146 and 251 to specifically indicate whether the evaluations were objective or subjective.

Explain in details the Randomisation - Was the process of selecting the topics to be tested on the generative AI-based model randomized?

The process of selecting the topics to be tested on the generative AI-based model was not randomised because that was not how our study was designed. That said, our original methods section did provide information about randomisation of the order of the MCQ questions (line 233-234).

Specificity is not clear, please clarify the following points:

How specific are the exact prompts used?

The specific prompts are already clearly described in Table 1. However, we have added a sentence to clarify the prompts’ specificity to sports nutrition (lines 93 and 221).

Were those exact prompts provided fully?

Yes. The specific prompts are clearly described in Table 1.

Did the authors consider the feedback and learning loops?

Yes, as described in the Prompt inputting sections (lines 115 and 226), new chatbot accounts were created, the chatbots’ histories were deleted before each new prompt was entered, the chatbots’ memories were turned off, and a new chat was created each time a new prompt was entered. We used these practices to prevent feedback and learning loop effects; we have added a sentence to clarify that on lines 121 and 240.

How specific are the language and cultural issues considered in the generative AI model?

All prompts were written in English (added text on lines 94 and 221). We added a sentence about cultural issues on line 112.

Discussion: The discussion section needs to be described scientifically. Kindly frame it along the following lines: i. Main findings of the present study. ii. Comparison with other studies. iii. Implication and explanation of findings. iv. Strengths and limitations. v. Conclusion, recommendation.

These are general comments on how to write a discussion section. Our original discussion already contains all these aspects.

Reviewer 3

Thank you for your comments. We have addressed them point-by-point in bolded blue font below.

1. Title, Abstract and Introduction

• Lines 8 – 14 lacks specific details about the statistical methods used to assess chatbot performance. Also, statistical methods for assessing interrater reliability and chatbot accuracy should be mentioned.

Thank you for spotting this omission. We have added text to address this point on lines 9 and 13.

• Lines 24 – 25 could be extended to include suggestions or recommendations for improving chatbot reliability.

Because the study was not designed to improve chatbot reliability in future versions, adding suggestions or recommendations for improving chatbot reliability would be an inappropriate concluding remark. So, we choose not to add such info.

• In lines 50 – 58, chatbot variability was mentioned but it does not explicitly discuss potential biases in AI-generated responses. It is important to address that.

An excellent point that we have now addressed on line 49.

• Details on how prompts were standardized across chatbots (lines 72-74) would improve replicability.

This information is already provided on lines 116-129.

• Potential biases in chatbot outputs (lines 75-76) should be explicitly acknowledged.

In line with your earlier comment, we added some text to line 49 in the introduction and further text in the limitations section of the discussion (line 650-654).

• Recommendations for improving chatbot-generated sports nutrition advice (lines 108 - 110) should be included.

While chatbot manufacturers should optimise their large language models by training them on the most recent sports nutrition guidelines and position statements, this study was not designed to make recommendations for improving chatbot-generated sports nutrition advice. Instead, the study was designed to compare the accuracy of nutritional knowledge between different chatbots. Therefore, adding explicit recommendations is inappropriate, and we refrain from doing so. That said, one aspect of our study was to determine how chatbot-generated outputs could be improved by using a detailed prompt containing best practice prompting techniques, including a context, a role, a specific chain of thought task, and notes to prevent lost-in-the-middle effects (e.g., Table 1 and line 98-100).

2. Study Design (Lines 80 – 83)

• It would be beneficial to explicitly mention any potential biases introduced by modifications to the original protocol. Clearly state what amendments were made post-registration and their potential impact on findings.

Lines 337-341 describe the amendments made post-registration. In brief, the amendments meant that Microsoft Copilot was never used in the study. This did not introduce any bias and does not impact the findings further th

---

## [Decision Letter · Decision Letter 1]

12 May 2025

Dear Dr. Laye,

Thank you for submitting your manuscript to PLOS ONE. After careful consideration, we feel that it has merit but does not fully meet PLOS ONE’s publication criteria as it currently stands. Therefore, we invite you to submit a revised version of the manuscript that addresses the points raised during the review process.

We look forward to receiving your revised manuscript.

Kind regards,

Sirwan Khalid Ahmed

Academic Editor

PLOS ONE

Journal Requirements:

Reviewers' comments:

Reviewer's Responses to Questions

**Comments to the Author**

Reviewer #1: All comments have been addressed

Reviewer #2: All comments have been addressed

Reviewer #3: All comments have been addressed

2. Is the manuscript technically sound, and do the data support the conclusions?

Reviewer #1: Yes

Reviewer #2: Yes

Reviewer #3: Yes

3. Has the statistical analysis been performed appropriately and rigorously?

Reviewer #1: Yes

Reviewer #2: Yes

Reviewer #3: Yes

4. Have the authors made all data underlying the findings in their manuscript fully available?

Reviewer #1: Yes

Reviewer #2: Yes

Reviewer #3: Yes

5. Is the manuscript presented in an intelligible fashion and written in standard English?

Reviewer #1: Yes

Reviewer #2: Yes

Reviewer #3: Yes

Reviewer #1: 1)The authors attribute the chatbots’ poor performance in certain areas (e.g., hydration, energy availability) mainly to general causes such as “training data bias.” However, this explanation should be expanded with more concrete examples and supported by references from the literature.

Furthermore, the manuscript would benefit from including practical suggestions for developers of AI systems. For example, it could recommend fine-tuning based on official sports nutrition guidelines. Currently, the recommendation to consult a dietitian is confined to a single sentence in the conclusion and does not sufficiently address the broader developmental implications.

2)While the distinctions between “simple” and “detailed” prompts are well defined, the manuscript does not adequately explain why some chatbots (e.g., Claude) responded more effectively to detailed prompts. A more in-depth discussion of how different models process prompt complexity would improve the interpretation of results.

3)Although the exclusion of variables such as gender and ethnicity from the prompts is methodologically explained, the possible consequences of this decision are not addressed in the discussion. The authors should discuss how the lack of gender- or culture-specific prompts might have influenced the outputs, especially considering that nutritional needs can vary across these dimensions.

Reviewer #2: THANKS ALOT FOR YOUR REPLAY

THANKS ALOT FOR YOUR REPLAY

THANKS ALOT FOR YOUR REPLAY

THANKS ALOT FOR YOUR REPLAY

THANKS ALOT FOR YOUR REPLAY

THANKS ALOT FOR YOUR REPLAY THANKS ALOT FOR YOUR REPLAY

Reviewer #3: (No Response)

**Do you want your identity to be public for this peer review?** For information about this choice, including consent withdrawal, please see our Privacy Policy

Reviewer #1: **Yes: ** Esedullah AKARAS

Reviewer #2: No

Reviewer #3: **Yes: ** Ojore Godday Aghedo

---

## [Author Response · Author response to Decision Letter 2]

20 May 2025

Please find our responses to reviewers attached as well as below.

Response to Reviewers

Reviewers' comments:

Reviewer 1

Thank you for your comments. We have addressed them point-by-point in bold blue font below.

1. The authors attribute the chatbots’ poor performance in certain areas (e.g., hydration, energy availability) mainly to general causes such as “training data bias.” However, this explanation should be expanded with more concrete examples and supported by references from the literature.

Furthermore, the manuscript would benefit from including practical suggestions for developers of AI systems. For example, it could recommend fine-tuning based on official sports nutrition guidelines. Currently, the recommendation to consult a dietitian is confined to a single sentence in the conclusion and does not sufficiently address the broader developmental implications.

We thank Reviewer #1 for highlighting the need for greater specificity and practical implications regarding chatbot limitations. To address this, we expanded our explanation beyond general terms like “training data bias” and have added some text to lines 600-602.

2. While the distinctions between “simple” and “detailed” prompts are well defined, the manuscript does not adequately explain why some chatbots (e.g., Claude) responded more effectively to detailed prompts. A more in-depth discussion of how different models process prompt complexity would improve the interpretation of results.

Thank you for this comment. We already discussed this point on lines 622-624 but have included some additional detail (lines 624-630) to further explain the differences in chatbot performance based on prompt type.

3. Although the exclusion of variables such as gender and ethnicity from the prompts is methodologically explained, the possible consequences of this decision are not addressed in the discussion. The authors should discuss how the lack of gender- or culture-specific prompts might have influenced the outputs, especially considering that nutritional needs can vary across these dimensions.

Thank you for this comment and insight.

Including demographic information such as gender, ethnicity, or cultural background in our prompts might have elicited more tailored and potentially more effective sports nutrition guidance. However, because the majority of sports nutrition research has been conducted on a relatively homogeneous population (often young white males), chatbot responses would likely have remained generalized or simply reflected the prevailing norms of that population (Turner et al., 2022; Cowley et al., 2022). Furthermore, as we already mentioned on lines 109-113, current position stands and consensus statements do not provide sports nutrition guidelines specific to race, ethnicity, culture, or gender. As a result, including demographic cues may not meaningfully improve personalization and could, in fact, have inadvertently reinforced existing biases. This concern is supported by recent evidence showing that large language models can generate biased clinical recommendations when demographic information is included in prompts (Sanghavi et al., 2024 https://www.nature.com/articles/s41591-025-03626-6#Sec9). Although this is an important issue, it is a general topic related to AI and sports nutrition ethics and is beyond the scope of the aim of our study. Therefore, we choose not to add any further text to the manuscript.

---

## [Editor Report · Decision Letter 2]

22 May 2025

The sports nutrition knowledge of large language model (LLM) artificial intelligence (AI) chatbots: an assessment of accuracy, completeness, clarity, quality of evidence, and test-retest reliability.

PONE-D-25-07476R2

Dear Dr. Laye,

We’re pleased to inform you that your manuscript has been judged scientifically suitable for publication and will be formally accepted for publication once it meets all outstanding technical requirements.

Kind regards,

Sirwan Khalid Ahmed

Academic Editor

PLOS ONE
---

## [Editor Report · Acceptance letter]

PONE-D-25-07476R2

PLOS ONE

Dear Dr. Laye,

I'm pleased to inform you that your manuscript has been deemed suitable for publication in PLOS ONE. Congratulations! Your manuscript is now being handed over to our production team.

Kind regards,

on behalf of

Dr. Sirwan Khalid Ahmed

Academic Editor

PLOS ONE